# Revisiting Data Challenges of Computational Pathology: A Pack-based Multiple Instance Learning Framework

## Abstract

Computational pathology (CPath) digitizes pathology slides into whole slide images (WSIs), enabling analysis for critical healthcare tasks such as cancer diagnosis and prognosis. However, WSIs possess extremely long sequence lengths (up to 200K), significant length variations (from 200 to 200K), and limited supervision. These extreme variations in sequence length lead to high data heterogeneity and redundancy. Conventional methods often compromise on training efficiency and optimization to preserve such heterogeneity under limited supervision. To comprehensively address these challenges, we propose a pack-based MIL framework. It packs multiple sampled, variable-length feature sequences into fixed-length ones, enabling batched training while preserving data heterogeneity. Moreover, we introduce a residual branch that composes discarded features from multiple slides into a *hyperslide* which is trained with tailored labels. It offers multi-slide supervision while mitigating feature loss from sampling. Meanwhile, an attention-driven downsampler is introduced to compress features in both branches to reduce redundancy. By alleviating these challenges, our approach achieves an accuracy improvement of up to 8% while using only 12% of the training time in the PANDA(UNI). Extensive experiments demonstrate that focusing data challenges in CPath holds significant potential in the era of foundation models. The code is here.

## 1 Introduction

Computational pathology (CPath) Song et al. (2023); Cifci et al. (2023) represents a rapidly evolving interdisciplinary research domain that integrates advanced computer vision techniques and pathology to facilitate accurate and efficient interpretation of histopathological images. Central to CPath are whole slide images (WSIs, slides), digitized pathology slides with gigapixel resolution often exceeding billions of pixels. It enables comprehensive microscopic analysis to support critical healthcare tasks such as cancer sub-typing Ilse et al. (2018); Zhang et al. (2022); Tu et al. (2022), grading Bulten et al. (2022), and prognosis Wen et al. (2023); Yao et al. (2020). Patching strategies help researchers effectively process these gigapixel images within hardware constraints. However, patches derived from WSIs present following challenges: as shown in Fig.1, **1)** they possess *extremely long sequence lengths* (up to 200K) and *significant sequence length variations* (from 200 to 200K). Such extreme distributions in sequence length, coupled with *diverse morphological characteristics*, contribute to *data heterogeneity*, which is substantial for CPath tasks. **2)** and introduce input *redundancy challenges* for CPath algorithms. **3)** Moreover, due to the gigapixel resolution and specialization, WSIs typically have *only slide-level annotations*, lacking more supervision that matches the complex input.

Current two-stage multiple instance learning (MIL) Maron & Lozano-Pérez (1997) paradigm Lu et al. (2021) is a compromise resulting from high data heterogeneity and limited supervision. This paradigm employs a pre-trained encoder to extract offline patch (instance) features, and uses a MIL model to produce slide-level (bag-level) results. Due to data challenges, it suffers from training inefficiency and instability. Specifically, with significant variations in sequence length across slides, mainstream methods typically process data with a batchsize of 1 during training Shao et al. (2021); Li et al. (2024c). While these approaches preserve whole-slide heterogeneity, training with a batchsize of 1 is inefficient (e.g., training TransMIL Shao et al. (2021) on the PANDA Bulten et al. (2022) dataset requires over 50 RTX3090 GPU-hours) and may yield suboptimal performance Koga et al.

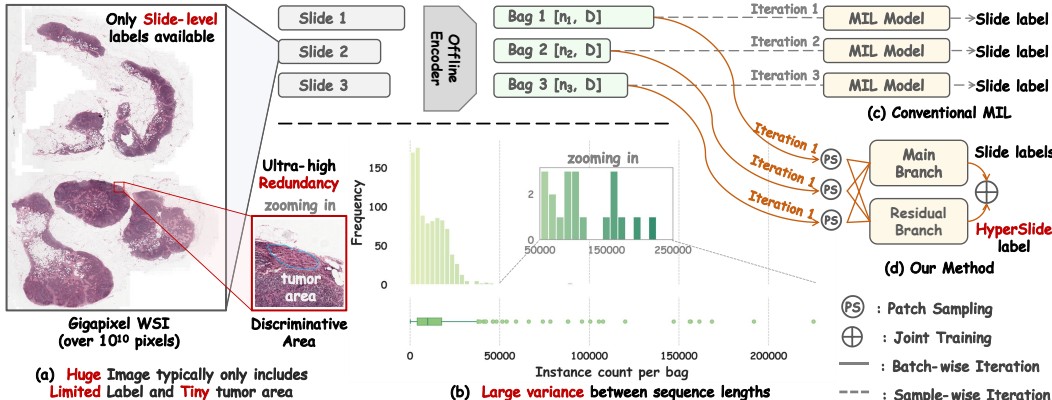

Figure 1: **(a, b)**: WSIs present significant data challenges, including high heterogeneity stemming from highly variable sequence lengths and diverse morphology, massive data redundancy, and limited supervision. **(c)**: Conventional methods train with batchsize of 1 to preserve data heterogeneity, suffering from training inefficiency and instability. **(d)**: Our pack-based framework packs variable-length sequences to preserve scale information. It further introduces a residual branch to model inter-slide correlations, constructing a *hyperslide* that retains all morphological features and enrich limited supervision. This approach maintains data heterogeneity while enabling batched training.

(2025). A few methods Campanella et al. (2019); Liu et al. (2024b) attempt to enable batched training by sampling or padding all sequences to a uniform length; however, this approach can lead to a loss of data heterogeneity and important features, especially affecting complex methods and tasks.

To comprehensively address three data challenges, we propose a novel pack-based MIL framework. Inspired by recent advancements in large model Pouransari et al. (2024); Krell et al. (2021); Dehghani et al. (2023); Wang et al. (2024a), it packs multiple variable-length sequences into a single fixed-length sequence to enable batched training while preserving data heterogeneity. However, leveraging packing strategies for effective batched training in CPath is far from straightforward. The excessive length of packed sequences hinders training, necessitating patch sampling, which still leads to feature loss. Therefore, we split the input features into main and residual branches, packing the retained and discarded features, respectively, to minimize sampling-induced feature loss. In the main branch, we employ masks to maintain the independence of different slides within a pack. Conversely, the residual branch treats discarded features from multiple slides in the same pack as a single *hyperslide*. To train this *hyperslide* effectively, we introduce task-specific hyperslide labels and loss functions. Crucially, this approach effectively offers multi-slide supervision.

While some outstanding works have explored supplementary supervision Zhang et al. (2022); Shao et al. (2023); Brussee et al. (2025); Fang et al. (2024), most focus on intra-slide modeling (e.g., instance-level or pseudo-bag), neglecting inter-slide relationships. Pathology slides from the same spatial and tissue origin exhibit consistent morphological characteristics Lin et al. (2025); Chen et al. (2022a); Kaczmarzyk et al. (2024). Learning inter-slide correlations allows the *hyperslide* to provide the model with a more comprehensive perspective, enabling the discovery of more generalizable pathological features. Furthermore, we propose an attention-driven downsampler to compress features for reducing input redundancy within both branches. To validate our framework, we conducted extensive experiments using features from foundation models. Results demonstrate that our approach consistently improves multiple baselines by effectively mitigating the data challenges inherent in CPath. Specifically, it delivers substantial performance gains (e.g., +11% accuracy on PANDA) while improving training efficiency ($\sim 8\times$ speedup on PANDA). Our contributions are:

- We revisit the data challenges in CPath, like high heterogeneity, high redundancy, and limited supervision. Considering these challenges, we propose an efficient and effective pack-based MIL framework that enables reliable training while preserving data heterogeneity.

- We construct the *hyperslide* from discarded features during the packing. Corresponding task-specific hyperslide labels and loss functions are designed. This strategy not only minimizes sampling-induced feature loss but also introduces multi-slide supervision. It provides the model with a more comprehensive perspective, thereby improving CPath performance.

- We propose an attention-driven downsampler to compress redundant features during the training process. With extensive experiments, we validate the effectiveness of the proposed approach, summarize practical guidelines for batched CPath training, and highlight the significant potential of addressing data challenges in the era of FM.

## 2 RELATED WORKS

**Supervision in Computational Pathology.** Recent CPath advancements leverage MIL to reduce annotation burden. Using only slide labels, MIL has evolved with mechanisms like attention Ilse et al. (2018); Tang et al. (2023), clustering Lin et al. (2023), Transformers Shao et al. (2021), and GNNs Wang et al. (2021); Eastwood et al. (2023) to improve interpretability, localization, and accuracy. Complementing pure MIL methods, pseudo-labeling strategies have emerged as powerful auxiliary techniques, encompassing instance-level pseudo-labeling Qu et al. (2022a), knowledge distillation frameworks Zhang et al. (2022); Qu et al. (2022b), limited pathologist patch annotations Koga et al. (2025), weak regional annotations Wang et al. (2022), and semi-supervised consistency regularization Jiang et al. (2023). These hybrid strategies effectively generate additional supervision to refine instance predictions and leverage unlabeled data, boosting performance and data efficiency. While some studies Liu et al. (2024a); Ouyang et al. (2024) explore mixup-like data augmentation between WSI pairs, supervision leveraging relationships across multiple slides is still largely unexplored.

**Batchsize in Computational Pathology.** Batchsize is a crucial hyperparameter in deep learning. However, its exploration in CPath remains limited, primarily due to the computational demands of gigapixel WSIs and the inherent data heterogeneity within each slide. Consequently, mainstream slide-level MIL methods typically adopt a batchsize of 1 Jaume et al. (2024); Li et al. (2024a); Shi et al. (2024); Li et al. (2024c); Song et al. (2024); Zhang et al. (2024b); Li et al. (2024b); Fourkioti et al. (2023). For instance, RRTMIL Tang et al. (2024) utilizes slide-wise regional and cross-region self-attention to capture patch ordinality and heterogeneity within each slide, necessitating a batchsize of 1 to maintain intra-slide relationships. Despite its prevalence, this practice often results in training instability and slow convergence, prompting methods such as gradient accumulation over multiple slides Koga et al. (2025); Zhang et al. (2025) or instance-level sampling strategies that select fixed-size subsets of patches per slide to mitigate computational overhead and improve learning stability Campanella et al. (2019); Liu et al. (2024b). Current slide-level methods treat batched inputs as an efficiency trade-off rather than a genuinely effective training strategy. In this paper, we investigate the benefits of batched training for slide-level prediction and explore practical guidelines for its implementation in CPath. Appendix E gives more discussion.

## 3 METHOD

### 3.1 PRELIMINARY: MIL-BASED COMPUTATIONAL PATHOLOGY

Histopathological WSIs are often gigapixel resolution, making direct processing impractical. Current approaches typically use weakly supervised MIL, where a WSI is treated as a bag $\mathcal{B} = \{x_1, \ldots, x_N\}$ of instances. During training, only a slide-level label $y$ is available. Each instance $x_i$ is encoded to an embedding $h_i = f(x_i)$ using a offline feature extractor $f$. An aggregation function $\Gamma(\cdot)$ combines

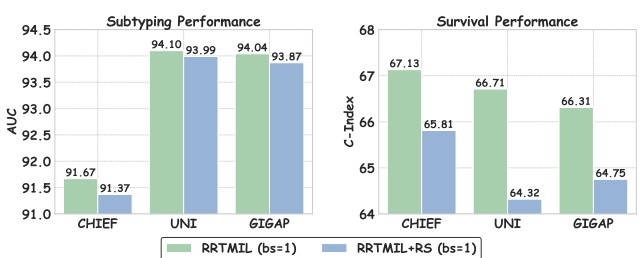

Figure 2: Impact of data heterogeneity on CPath.

instance embeddings $\{h_i\}_{i=1}^N$ into a bag-level representation $z = \Gamma_\theta(\sum_{i=1}^N h_i)$. This representation $z$ is then used by a classifier $g_\phi$ to predict the slide label $p(y \mid \mathcal{B}) = g_\phi(z)$. The significant variation in instance count $N$ across WSIs contribute to data heterogeneity. This heterogeneity along with its associated spatial and morphological context, is crucial for CPath, as shown in Fig.2. Specifically, when all instances are randomly sampled to a fixed-length sequence, a latest method like RRTMIL Tang et al. (2024) exhibits consistent performance degradation on multiple benchmarks, particularly for

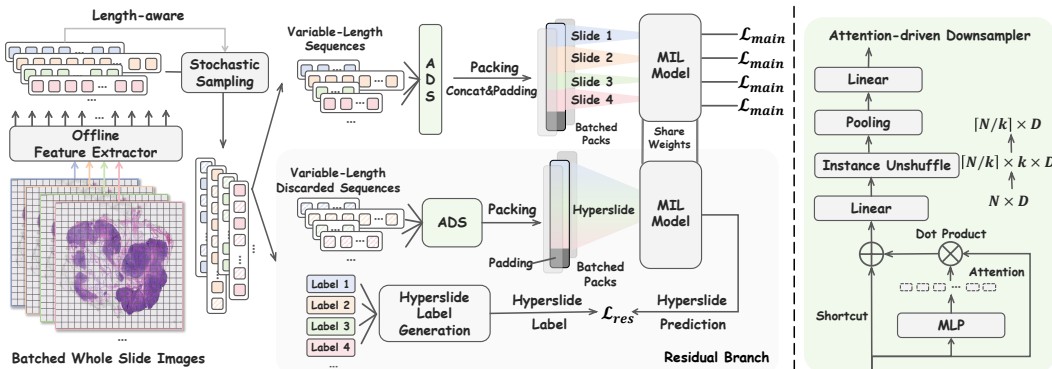

Figure 3: **Left:** Overview of the proposed pack-based MIL training framework. **Right:** llustration of Attention-driven Downsampler (ADS).

complex tasks like survival analysis. To maintain data heterogeneity, current methods necessitates training with a batchsize of 1, resulting in noisy gradient estimates and optimization instability.

## 3.2 PACK-BASED MIL FRAMEWORK

CPath poses significant data challenges, including high data heterogeneity, redundancy and limited supervision, which hinder mainstream MIL method. To overcome these issues, we propose a pack-based MIL framework, named PackMIL. As illustrated in Fig.3, the proposed method effectively employs a packing operation to maintain data heterogeneity and allow batched input, and it further provides multi-slide supervision, leading to more effective and stable training. Consider a mini-batch of $B$ bags, the $b$-th bag provides a set of instance embeddings $\mathcal{B}_b = \{h_{bi}\}_{i=1}^{N_b}$ where $h_{bi} \in \mathbb{R}^D$. However, the feature sequences extracted from gigapixel WSIs can be prohibitively long, making their direct inclusion into fixed-length packs impractical due to memory and computational limitations during training. To address this, we employ stochastic instance-level sampling with ratio $r \in (0, 1)$. The subsets $\mathcal{R}_b$ and $\mathcal{D}_b$ are formed as:

$$m_{bi} \sim \text{Bernoulli}(1 - r), \quad \mathcal{R}_b = \{h_{bi} \mid m_{bi} = 1\}, \quad \mathcal{D}_b = \{h_{bi} \mid m_{bi} = 0\}, \quad (1)$$

We introduce the Residual Branch to mitigate feature loss during sampling and establish hyperslide supervision. The retained set $\mathcal{R}$ is forwarded in main branch, while discarded set $\mathcal{D}$ is forwarded in residual branch. For each bag $b$, the corresponding sets $\mathcal{R}_b$ and $\mathcal{D}_b$ are processed to produce downsampled feature sets, denoted as $\tilde{\mathcal{R}}_b$ and $\tilde{\mathcal{D}}_b$, respectively. To maintain data heterogeneity for batch training, instances from sampled sets of each bag are sequentially arranged into fixed-length packs of size $L$. This packing operation $\text{PACK}(\cdot)$ processes all bags in the mini-batch to form:

$$P^{\text{main}} = \text{PACK}\Big(\bigcup_{b=1}^{B} \tilde{\mathcal{R}}_b, \, L\Big) \in \mathbb{R}^{B' \times L \times D}, P^{\text{res}} = \text{PACK}\Big(\bigcup_{b=1}^{B} \tilde{\mathcal{D}}_b, \, L\Big) \in \mathbb{R}^{B'' \times L \times D}, \quad (2)$$

where $\cup$ denotes concatenation along the instance axis and $B'$ and $B''$ are the number of packs generated for the main and residual branches, respectively, calculated from the number of instances and $L$. To ensure adequate representation of each slide, we enforce a minimum number of patches per slide in each pack. Zero-padding is applied in each pack when it contains fewer than $L$ patches.

We also incorporate an attention-driven downsampler (ADS) module for handling input redundancy between the two branches. This module fuses features from instances in $\mathcal{R}_b$ and $\mathcal{D}_b$ to generate more compact and informative feature representations. Each set $\mathcal{S} \in \{\mathcal{R}_b, \mathcal{D}_b\}$ is downsampled by a factor $k \in \mathbb{N}$ using an ADS module, yielding $\tilde{\mathcal{S}} = \text{ADS}(\mathcal{S}; k)$ with size $|\tilde{\mathcal{S}}| = \lceil |\mathcal{S}|/k \rceil$.

After packing, the main and residual branches are processed independently while sharing the same MIL model weight. In the main branch, each pack contains instances originating from $B$ distinct bags. We use a mask $\mathbf{M}_p$ to identify the source bag for each instance within pack $p$. The embedding for the $b$-th bag is then computed by selectively aggregating its instances from the pack: $z_b^{\text{main}} = \Gamma_\theta\Big(\mathbf{M}_{pb}, P_p^{\text{main}}\Big)$. We compute the main loss $\mathcal{L}_{\text{main}}$ over $B$ slides, based on slide-level predictions

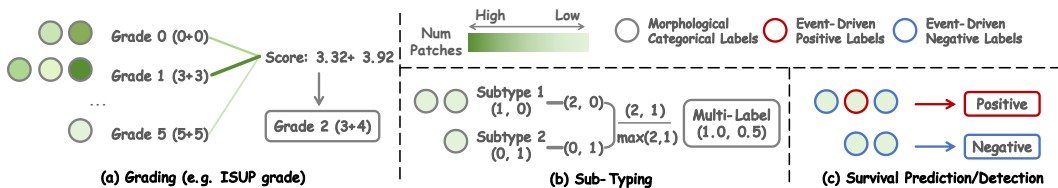

Figure 4: Illustration of Task-specific Hyperslide Labels.

$\hat{y}_b = g_\phi(z_b^{\text{main}})$. For the residual branch, each pack $p$ acts as a hyperslide, introducing high-level supervision. We compute its embedding $z_p^{\text{res}} = \Gamma_\theta\left(P_p^{\text{res}}\right)$, obtain pack-level predictions $\hat{y}_p^{\text{hyper}} = g_\phi(z_p^{\text{res}})$, and compute the residual loss $\mathcal{L}_{\text{res}}$ over $B''$ packs. The overall objective function is a weighted sum of the two losses: $\mathcal{L} = \mathcal{L}_{\text{main}} + \lambda\,\mathcal{L}_{\text{res}}$.

**Packing.** The packing operation processes concatenated downsampled embeddings, denoted as $\mathcal{H} = \bigcup_{b=1}^{B} \tilde{\mathcal{R}}_b = \{h_n\}_{n=1}^{M}$, where $M$ is total number of features in the mini-batch and $\beta_n$ is original bag index of feature $h_n$. The operation sequentially fills packs of fixed length $L$. Let $P_p \in \mathbb{R}^{L \times D}$ be the $p$-th pack, for $p = 1, \ldots, B'$. Features $\{h_n\}_{n=1}^{M}$ are placed sequentially into $P_1, P_2, \ldots, P_{B'}$. When a pack $P_p$ cannot accommodate the next feature without exceeding $L$ tokens, or when all $M$ features have been placed, any remaining positions in $P_p$ are filled with vectors $\mathbf{0} \in \mathbb{R}^D$.

**Isolated Mask.** We employ auxiliary masks to process features within each pack $P_p$ while preserving original bag integrity. Based on the CPath pipeline above, masks are divided into *aggregation-oriented* and *classification-oriented*. Aggregation masks constrain the feature aggregation stages, ensuring that computations within each pack only involve instances from the corresponding source bag. For example, in Transformer-based models like TransMIL Shao et al. (2021), this mask restricts Multi-Head Self-Attention computations strictly to features originating from the same source bag within the pack, preventing cross-bag attention and aggregation. Classification-oriented masks subsequently select tokens relevant for prediction. This mask identifies source bag for each non-padding token, enabling appropriate classification based on bag identity. These tailored masks enable efficient and effective MIL within our pack-based framework. Detailed definition is provided in Appendix D.2.

**Attention-driven Downsampler (ADS).** WSIs exhibit significant redundancy Tang et al. (2023); Zhang et al. (2024b), which can also manifest between the feature sets derived from $\mathcal{R}_b$ and $\mathcal{D}_b$. To fuse these features while mitigating redundancy in dual branches, we designed the ADS module. This module utilizes attention-driven downsampling to produce a compact and informative representation. Given a set of $N$ embeddings $\{h_i\}_{i=1}^{N}$, ADS first computes a per-instance attention score via a shallow MLP and applies it as a residual weigh $u_i = h_i + a_i\,h_i$. Followed by a learnable linear layer, we compute $W^L \in \mathbb{R}^{D \times D}$, $v_i = u_i W^L$. To facilitate structured downsampling that preserves spatial relationships, we perform instance unshuffle, which rearranges the sequential features into a group-based representation:

$$[v_1, \ldots, v_N] \in \mathbb{R}^{N \times D} \xrightarrow{\text{unshuffle by } k} U \in \mathbb{R}^{\lceil N/k \rceil \times k \times D}. \tag{3}$$

where $k$ is the downsampling factor. ADS poolings along the $k$ dimension of each pack, followed by a projection $W^P \in \mathbb{R}^{D \times D}$. ADS takes the input features $\{h_i\}$ and a factor $k$, resulting in the output sequence $\text{ADS}(\{h_i\}; k) = \{\tilde{h}_j\}_{j=1}^{\lceil N/k \rceil} \in \mathbb{R}^{\lceil N/k \rceil \times D}$. Each downsampled feature $\tilde{h}_j$ is computed as:

$$\tilde{h}_j = \left[\text{Pool}(U_j, \dim = 1)\right] W^P, \quad j = 1, \ldots, \lceil N/k \rceil. \tag{4}$$

where $\text{Pool}(\cdot)$ is either random or max pooling, and $W^P \in \mathbb{R}^{D \times D}$ is a projection matrix. By weighting instances with attention scores $a_i$ and pooling across structured groups, ADS reduces the instance count by a factor of $k$ while prioritizing regions of high clinical relevance. ADS also maintains interpretability at inference time. Additional details and a discussion on the applicable boundaries of ADS are provided in Appendix B.3.

**Inference Pipeline.** For inference, we adopt a deterministic pipeline to ensure reproducible predictions. The residual branch and stochastic sampling are bypassed. Each slide is processed individually with batchsize of 1, feeding its complete sequence of instance embeddings into the main branch.

## 3.3 TRAINING RECIPE

**Task-specific Hyperslide Labels and Loss.** Heterogeneous pathological data and diverse downstream clinical tasks necessitate specialized label generation strategies that preserve task-relevant clinical characteristics and enable efficient learning from multiple WSIs, encouraging the design of task-specific hyperslide labeling mechanisms. We divide downstream tasks into two classes: *Morphological Categorical* and *Event-Driven*. Based on pathological scenarios, we design three strategies to generate a hyperslide label $y^{\text{hyper}}$ for higher-level supervision, illustrated in Fig.4. Appropriate loss functions are selected for different tasks, as detailed in the Appendix D.1.

1) Grading. Let each pack contain $S$ WSIs with pathologist-assigned scores or tumour-area ratios $g_s$ and corresponding patch counts $n_s$. We compute a pack-level statistic by $\tilde{g} = \frac{\sum_{s=1}^{S} n_s\, g_s}{\sum_{s=1}^{S} n_s}$. The weighted score (or ratio) is then mapped to the discrete grading rubric recommended by the pathology guideline, yielding a single categorical hyperslide label $y^{\text{hyper}} \in \{1, \ldots, G\}$. Weighting by $n_s$ ensures that the pack-level grading reflects the relative tissue coverage of each WSI, preventing smaller tissue regions from being overrepresented in the final assessment.

2) Sub-typing. Sub-typing is cast as a *multi-label* problem in which a single pack may express several subtypes simultaneously. For each subtype $c \in \{1, \ldots, C\}$ we first obtain a slide-level indicator $t_{s,c} \in \{0, 1\}$ and aggregate groundtruth across the $S$ slides by $\hat{p}_c = \frac{\sum_{s=1}^{S} n_s\, t_{s,c}}{\sum_{s=1}^{S} n_s}$, where $n_s$ denotes the patch count of slide $s$. To capture the relative prevalence of co-occurring subtypes within pack, we generate hyperslide soft label $\mathbf{y}^{\text{hyper}}$ by normalizing aggregated values using maximum value across all subtypes: $y_c^{\text{hyper}} = \frac{\hat{p}_c}{\max_{j \in \{1, \ldots, C\}} \hat{p}_j}$, resulting in $\mathbf{y}^{\text{hyper}} = [y_1^{\text{hyper}}, \ldots, y_C^{\text{hyper}}]^{\top} \in [0, 1]^C$.

3) Survival Analysis and Detection. These tasks are inherently event-driven: labels correspond to clinical events with an intrinsic priority (e.g., tumor overrides normal). We preserve this hierarchy by scanning the slides in descending priority order and assigning the first event observed:

$$y^{\text{hyper}} = \arg\max_{e \in \mathcal{E}} \Big[ \max_{s} \mathbf{1}\{ e \text{ occurs in slide } s \} \Big], \tag{5}$$

where $\mathcal{E}$ is the ordered event set. This strategy ensures that each bag is uniquely associated with the most clinically significant event, aligning label semantics closely with clinically practice.

**Practical Guidelines for Batched Training.** Selecting an appropriate batchsize ($bs$) across various CPath tasks remains an underexplored yet critical problem for effective model training. To establish an batched training protocol, we conduct a series of preliminary experiments. The results from these experiments inform the practical guidelines discussed below, which are subsequently adopted in our main experimental setup. We find that the specifics of these guidelines vary significantly across benchmarks and are strongly correlated with dataset scale.

On conventional CPath datasets (e.g., TCGA), which typically contain a limited number of gigapixel WSIs , resource constraints often force a trade-off between the $bs$ and the number of patches sampled per WSI (NP), as shown in the right figure(b). The optimal choice of $bs$ usually requires empirical tuning. Additionally, regular rules for scaling the learning rate do not apply and also necessitate empirical tuning, as depicted in the right figure(c).

On large-scale CPath datasets (e.g., PANDA Bulten et al. (2022)), the $\sqrt{bs}$ learning rate scaling rule often proves effective. However, performance does not monotonically increase with $bs$; excessively large values can degrade performance, as illustrated in the right figure(a). Moreover, we found that 1D Batch Normalization Ioffe & Szegedy (2015) can effectively facilitate convergence on benchmarks with sufficient data scale. In contrast, normalization does not provide significant improvements on conventional CPath datasets, which we attribute to their limited data volume.

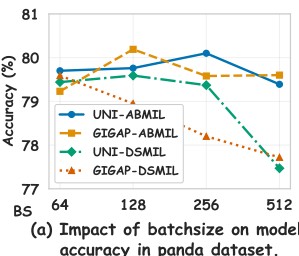

(a) Impact of batchsize on model accuracy in panda dataset.

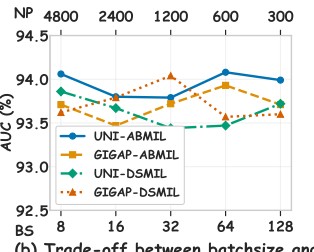

(b) Trade-off between batchsize and number of patches in TCGA dataset.

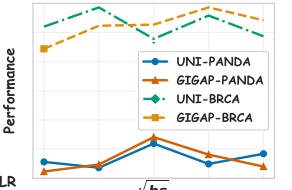

(c) Effect of LR on model performance in PANDA and TCGA dataset.

Table 1: Comparison of ISUP Grading (Acc.) on PANDA and Sub-typing (AUC) on BRCA.

| Method | Grading (Acc.↑) | | | Sub-typing (AUC↑) | | |
|---|---|---|---|---|---|---|
| | CHIEF (27M) | UNI (307M) | GIGAP (1134M) | CHIEF | UNI | GIGAP |
| ABMIL Ilse et al. (2018) | $65.48_{\pm 0.70}$ | $73.21_{\pm 0.61}$ | $72.91_{\pm 0.62}$ | $89.58_{\pm 5.15}$ | $93.58_{\pm 3.92}$ | $94.13_{\pm 3.88}$ |
| DSMIL Li et al. (2021) | $71.95_{\pm 0.72}$ | $71.41_{\pm 0.68}$ | $70.84_{\pm 0.73}$ | $90.68_{\pm 4.91}$ | $93.89_{\pm 3.79}$ | $93.70_{\pm 4.14}$ |
| CLAM Lu et al. (2021) | $66.91_{\pm 0.73}$ | $74.76_{\pm 0.61}$ | $74.99_{\pm 0.60}$ | $89.61_{\pm 5.05}$ | $93.90_{\pm 3.86}$ | $93.73_{\pm 3.75}$ |
| TransMIL Shao et al. (2021) | $68.53_{\pm 0.79}$ | $72.59_{\pm 0.75}$ | $71.91_{\pm 0.74}$ | $91.91_{\pm 4.34}$ | $94.09_{\pm 3.79}$ | $93.57_{\pm 3.84}$ |
| DTFD Zhang et al. (2022) | $63.57_{\pm 0.71}$ | $72.19_{\pm 0.66}$ | $71.93_{\pm 0.65}$ | $91.07_{\pm 4.59}$ | $93.86_{\pm 3.93}$ | $93.88_{\pm 3.81}$ |
| WiKG Li et al. (2024c) | $72.37_{\pm 0.72}$ | $76.68_{\pm 0.68}$ | $74.91_{\pm 0.68}$ | $91.93_{\pm 4.65}$ | $93.81_{\pm 4.25}$ | $94.06_{\pm 3.92}$ |
| RRTMIL Tang et al. (2024) | $69.99_{\pm 0.69}$ | $72.18_{\pm 0.67}$ | $72.34_{\pm 0.63}$ | $91.67_{\pm 4.77}$ | $94.10_{\pm 3.65}$ | $94.04_{\pm 3.86}$ |
| 2DMamba Zhang et al. (2024a) | $71.59_{\pm 0.73}$ | $74.97_{\pm 0.68}$ | $75.36_{\pm 0.66}$ | $90.96_{\pm 3.85}$ | $93.59_{\pm 3.14}$ | $93.33_{\pm 3.25}$ |
| ABMIL+RS | 74.72 +9.2 | 77.91 +4.7 | 78.97 +6.1 | 88.72 −0.9 | 93.89 +0.4 | 93.78 -0.4 |
| ABMIL+PackMIL (Ours) | **76.46 +11.0** | **80.19 +7.0** | **80.41 +7.5** | 92.38 +2.8 | **94.86 +1.3** | **94.86 +0.7** |
| DSMIL+RS | 75.00 +3.1 | 78.59 +7.2 | 78.60 +7.8 | 91.62 +0.9 | 93.29 −0.6 | 94.04 +0.3 |
| DSMIL+PackMIL (Ours) | 75.84 +3.9 | 79.68 +8.3 | 79.10 +8.3 | **93.01 +2.3** | 94.62 +0.7 | 94.65 +1.0 |
| TransMIL+RS | 73.68 +5.2 | 76.94 +4.4 | 76.15 +4.2 | 90.75 −1.2 | 94.07 −0.0 | 93.95 +0.4 |
| TransMIL+PackMIL (Ours) | 74.75 +6.2 | 78.87 +6.3 | 78.88 +7.0 | 92.31 +1.0 | 94.37 +0.3 | 94.12 +0.6 |
| RRTMIL+RS | 70.32 +0.3 | 75.04 +2.9 | 75.13 +2.8 | 91.55 −0.1 | 94.02 -0.1 | 93.70 -0.3 |
| RRTMIL+PackMIL (Ours) | 74.63 +4.6 | 78.46 +6.3 | 78.43 +6.1 | 92.43 +0.9 | 94.54 +0.4 | 94.47 +0.4 |

## 4 EXPERIMENT

### 4.1 DATASETS AND EVALUATION METRICS

For cancer diagnosis, we evaluate performance on cancer grading and sub-typing tasks using the **PANDA** Bulten et al. (2022) and **TCGA-BRCA** datasets. For cancer prognosis, we evaluate survival analysis performance using **TCGA-LUAD** and **TCGA-BRCA**. We report macro accuracy (Acc.) for cancer grading and the area under the ROC curve (AUC) for sub-typing. For survival analysis, we report the concordance index (C-index) Harrell Jr et al. (1996). To ensure statistical robustness, we perform 1000 bootstrap iterations and repeat experiments across multiple random seeds. We report the mean and 95% confidence interval for all metrics. Appendix A gives further details.

### 4.2 MAIN RESULTS

**Comparison Methods.** We compare several established and recent MIL aggregators Ilse et al. (2018); Lu et al. (2021); Shao et al. (2021); Li et al. (2021; 2024c); Tang et al. (2024); Zhang et al. (2024a; 2022) using three SOTA pathology encoders: UNI Chen et al. (2024), CHIEF Wang et al. (2024b), and GigaPath (GIGAP) Xu et al. (2024). To comprehensively evaluate the proposed framework, we select four widely-used MIL models as baselines. Additionally, we compare PackMIL against a standard random sampling (RS) strategy (i.e., sampling all inputs to a fixed length for batched training) to assess its effectiveness.

**Focusing on Data Challenges in the FM Era.** Although MIL architectures have advanced significantly, their performance improvements become marginal when using offline features extracted by foundation models (FMs). The quality of the FM primarily determines the final performance, and the latest or more complex MIL methods have reached a performance bottleneck. In this context, we observe that *addressing the inherent data challenges in CPath is an effective way to achieve significant performance gains.* Specifically, batched training based on random sampling (RS) achieves substantial improvements on grading tasks (Tab. 1), especially on large datasets such as PANDA (~10,000 slides). We attribute this primarily to the advantages of batched training on large-scale data and its effectiveness in reducing input redundancy. However, this RS strategy performs inconsistently on benchmarks like TCGA-BRCA-subtyping, which exhibits greater data heterogeneity (e.g., a sequence length variation of 60,000 compared to 1,000 in the PANDA dataset). It provides only marginal gains on some benchmarks while degrading performance on others in this task. As shown in Tab. 2, this issue becomes more pronounced in survival analysis, where sequence length variation are even larger. This sensitivity to heterogeneity is further reflected in its performance on complex methods like RRTMIL, where RS shows a significant performance gap across all benchmarks. These results indicate that RS compromises the WSI heterogeneity preserved during traditional training (batchsize = 1) and suffers from feature loss due to sampling.

Table 2: Comparison of Survival Analysis (C-index Harrell Jr et al. (1996)) on BRCA and LUAD. OOM denotes Out-of-Memory in 24GB-RTX3090.

| Method | Survival-BRCA (C-index↑) | | | Survival-LUAD (C-index↑) | | |
|---|---|---|---|---|---|---|
| | CHIEF (27M) | UNI (307M) | GIGAP (1134M) | CHIEF | UNI | GIGAP |
| ABMIL Ilse et al. (2018) | $65.36_{\pm 9.00}$ | $65.87_{\pm 9.43}$ | $65.72_{\pm 8.86}$ | $61.99_{\pm 8.70}$ | $60.54_{\pm 8.83}$ | $59.88_{\pm 8.60}$ |
| DSMIL Li et al. (2021) | $65.81_{\pm 9.17}$ | $65.87_{\pm 9.98}$ | $64.94_{\pm 9.22}$ | $62.38_{\pm 8.50}$ | $61.44_{\pm 8.76}$ | $61.35_{\pm 8.20}$ |
| CLAM Lu et al. (2021) | $65.03_{\pm 9.38}$ | $65.45_{\pm 10.0}$ | $63.91_{\pm 9.70}$ | $61.78_{\pm 8.84}$ | $59.71_{\pm 8.48}$ | $60.70_{\pm 8.68}$ |
| TransMIL Shao et al. (2021) | $65.75_{\pm 8.87}$ | $65.34_{\pm 9.44}$ | $65.35_{\pm 9.18}$ | $63.68_{\pm 8.66}$ | $62.63_{\pm 8.70}$ | $62.53_{\pm 8.53}$ |
| DTFD Zhang et al. (2022) | $67.22_{\pm 8.91}$ | $65.05_{\pm 10.3}$ | $65.54_{\pm 9.08}$ | $62.87_{\pm 8.56}$ | $60.83_{\pm 8.64}$ | $60.88_{\pm 8.67}$ |
| WIKG Li et al. (2024c) | $65.55_{\pm 9.14}$ | $65.77_{\pm 9.21}$ | $65.79_{\pm 9.62}$ | OOM | OOM | OOM |
| RRTMIL Tang et al. (2024) | $67.13_{\pm 8.77}$ | $66.71_{\pm 9.95}$ | $66.31_{\pm 9.48}$ | $63.51_{\pm 8.76}$ | $61.32_{\pm 8.73}$ | $62.41_{\pm 8.41}$ |
| Mamba2D Zhang et al. (2024a) | $66.01_{\pm 7.06}$ | $65.73_{\pm 8.15}$ | $65.96_{\pm 7.69}$ | $63.23_{\pm 6.70}$ | $60.69_{\pm 6.97}$ | $62.19_{\pm 6.73}$ |
| ABMIL+RS | 64.02 −1.3 | 65.71 −0.2 | 63.88 −1.8 | 61.98 -0.1 | 60.34 −0.2 | 61.01 +1.0 |
| ABMIL+PackMIL (Ours) | 68.30 +2.9 | 68.14 +2.3 | 67.04 +1.3 | 63.72 +1.8 | 62.60 +1.0 | 61.58 +1.7 |
| DSMIL+RS | 65.00 −0.8 | 64.72 −1.1 | 65.69 +0.8 | 63.28 +0.9 | 61.53 +0.1 | 61.00 −0.4 |
| DSMIL+PackMIL (Ours) | **69.76 +2.9** | **70.00 +4.1** | **68.03 +3.1** | 64.10 +1.7 | 62.18 +0.7 | 62.44 +1.1 |
| TransMIL+RS | 66.39 +0.6 | 65.14 −0.2 | 65.52 +0.2 | 63.53 −0.2 | 62.12 −0.5 | 61.03 −1.5 |
| TransMIL+PackMIL (Ours) | 68.08 +2.3 | 68.44 +3.1 | 66.80 +1.5 | 64.01 +0.3 | **63.61 +1.0** | **63.04 +0.5** |
| RRTMIL+RS | 66.11 −1.0 | 64.68 −2.0 | 65.19 −1.1 | 62.52 −1.0 | 61.44 +0.1 | 61.35 −1.1 |
| RRTMIL+PackMIL (Ours) | 68.15 +1.0 | 68.73 +2.0 | 67.62 +1.3 | **64.37 +0.9** | 62.01 +0.7 | 62.79 +0.4 |

**PackMIL More Comprehensively Alleviates Data Challenges.** Compared with RS, PackMIL more comprehensively alleviates the various data challenges in CPath, achieving more substantial and consistent performance gains. For example, its superior performance in survival analysis demonstrates its ability to handle data heterogeneity. Moreover, PackMIL yields significant further improvements even on grading tasks where RS already performed well. We attribute this gain to PackMIL's ability to mitigate challenges of insufficient supervision. Notably, by comprehensively addressing these data challenges, PackMIL enables models to achieve performance comparable to higher-quality features with lower-cost offline features. Furthermore, it achieves consistent improvements when paired with better features, demonstrating its generalizability. In summary, results demonstrate the significant impact of inherent data challenges on CPath performance and validate the effectiveness of PackMIL.

## 4.3 ABLATION STUDY

In this subsection, we systematically ablate the components of PackMIL. By default, we use ABMIL as the baseline model and UNI as the offline feature extractor. For the survival analysis task, we utilize the larger BRCA dataset. Appendix B gives extra discussions.

**Batched Training and Packing Strategy.** The results at the middle of Tab. 3 show that Random Sampling (RS) achieves significant improvements on the grading task, but its performance degrades on the other two tasks. This is attributed to its effective mitigation of input redundancy. However, this approach compromises data heterogeneity and leads to feature loss. Furthermore, batched training yields consistent performance improvements (row 7). Importantly, it leads to significant gains in training stability and efficiency. Specifically, while maintaining faster and more stable convergence (Tab. 3(a)), it also alleviates overfitting on the test set. With the pack-based batched training scheme, PackMIL retains data heterogeneity. This allows PackMIL to benefit from batch processing while preserving the performance advantages of traditional training (batchsize=1).

**Hyperslide.** We constructs a *hyperslide* using discarded features in the residual branch and task-relevant hyperslide labels, aiming to offer multi-slide supervision while mitigating feature loss from sampling. Tab. 3(b) confirms that the hyperslide learns effectively when guided by the proposed task-relevant labels. Furthermore, during joint training with the task loss, we find that optimizing the hyperslide also helps mitigate the rapid overfitting (i.e., the model rapid convergence of the training loss to near-zero on the training data within early epochs) of the model to the task loss on the FM features (Tab. 3(c)). This issue hinders the model from benefiting from the task loss, consequently reducing training quality. The results in row 8 and row 10 of Tab. 3 demonstrate the performance improvements achieved by multi-slide supervision and minimizing feature loss.

**Attention-driven Downsampler.** The Attention-driven Downsampler (ADS) is designed to mitigate potential input redundancy via feature fusion. Since the PANDA dataset contains fewer patches per WSI, we do not employ ADS on this task. However, for the sub-typing and survival tasks, which

Table 3: **Top:** Ablation of PackMIL and computational cost on PANDA. TTime (RTX 3090 GPU-hours) denotes Train Time. Memory is the GPU memory which evaluated during training. FPS stands for frames per second. ADS module is disabled by default on PANDA; we detail its computational cost in Appendix B.3. **Bottom:** Loss curves of main and hyperslide loss under different settings.

| Method | Batched Training | TTime | Memory | FPS | Grad. | Sub. | Surv. |
|---|---|---|---|---|---|---|---|
| ABMIL Ilse et al. (2018) | ✗ | 12h | 0.6G | 2056 | 73.21 | 93.58 | 65.87 |
| PackMIL(AB.) (Ours) | ✓ | 4h | 2.8G | 1984 | **80.19** | **94.86** | 68.14 |
| TransMIL Shao et al. (2021) | ✗ | 55h | 1.1G | 142 | 72.59 | 94.09 | 65.34 |
| PackMIL(Trans.) (Ours) | ✓ | 6.5h | 4.4G | 731 | 78.87 | 94.37 | **68.44** |
| ABMIL (Baseline) | ✗ | 12h | 0.6G | 2056 | 73.21 | 93.58 | 65.87 |
| + Random Sampling | ✗ | 12h | 0.5G | 2056 | 77.62 | 93.39 | 65.33 |
| + Random Sampling | ✓ | 2h | 1.3G | 2056 | 77.91 | 93.89 | 65.71 |
| + Random Sampling + HyperSlide | ✓ | 2.5h | 2.5G | 2056 | 79.93 | 94.37 | 67.15 |
| + Pack | ✓ | 2.5h | 2.2G | 1984 | 79.02 | 94.06 | 66.15 |
| + Pack + HyperSlide | ✓ | 4h | 2.8G | 1984 | 80.19 | 94.21 | 67.50 |
| + Pack + HyperSlide + ADS | ✓ | - | - | - | - | 94.86 | 68.14 |

*(a) Training loss convergence.* Batched training (*orange*) exhibits faster and more stable convergence compared to the baseline with a batchsize of one (*blue*). *(b) Auxiliary hyperslide loss.* The decreasing loss curve for the auxiliary hyperslide task demonstrates that the model effectively learns from the proposed multi-slide supervision. *(c) Impact of hyperslide on task loss.* The hyperslide (*orange*) acts as a regularizer, mitigating the rapid overfitting (i.e., loss approaching zero in early epochs) observed when training with the task loss alone (*blue*).

involve longer input sequences, the results indicate that ADS effectively reduces input redundancy. More important, through attention weighting and feature fusion, ADS preserves the discriminative information of the original features. Appendix B.3 gives further discussion.

**Training Cost Analysis.** On traditional small-scale datasets, CPath algorithms based on offline features rarely face significant efficiency challenges. However, this issue becomes increasingly prominent with the advent of large-scale datasets, such as PANDA. Due to the non-batched training, we observe that training even a simple baseline like ABMIL on PANDA using offline features consumes 12 RTX3090 GPU-hours, while a more complex model like TransMIL requires 55 GPU-hours (top of Tab. 3). These results highlight how non-batched training severely hinders the scalability of CPath algorithms to larger datasets. By enabling batched training, training time is significantly reduced (-6∼9x), and accompanied by performance improvements. Notably, PackMIL achieves further significant performance gains with only a minor increase in computational overhead. It still maintaining a substantial efficiency advantage over non-batched training (∼12% training time).

## 5 CONCLUSION

In CPath, gigapixel WSIs exhibit extremely long sequences, significant length variations, high redundancy, and limited supervision. Existing methods typically address only individual aspects of these challenges, lacking systematic exploration. Our work reveals that these challenges lead to training inefficiency, instability, and high redundancy on both large-scale and conventional datasets. To comprehensively tackle these issues, we propose the pack-based MIL framework. It enables batch training while preserving data heterogeneity, enhancing both training efficiency and quality by a large margin. We incorporates a residual branch that utilizes *hyperslides* to supplements the limited supervision while mitigating feature loss from packing. Moreover, a attention-driven downsampler is integrated to compress feature redundancy within both branches. We also systematically evaluated a simple random sampling training strategy, which demonstrated considerable improvements on the PANDA. With extensive experiments, we summarize practical guidelines for batched CPath training and highlight the significant potential of focusing on data challenges in the era of FM.

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

# Appendix

# Table of Contents

# A  DATASETS AND IMPLEMENTATION DETAILS

## A.1  DATASETS

We validate our method on various computational pathology tasks and challenging benchmarks in the ear of foundation models, including cancer grading (PANDA Bulten et al. (2022)), subtyping (TCGA-BRCA), survival analysis (TCGA-LUAD, TCGA-BRCA).

**PANDA** Bulten et al. (2022) is a large-scale, multi-center dataset dedicated to prostate cancer detection and grading. It comprises 10,202 digitized H&E-stained whole-slide images, making it one of the most extensive public resources for prostate cancer histopathology. Each slide is annotated according to the Gleason grading system and subsequently assigned an International Society of Urological Pathology (ISUP) grade, enabling both cancer detection and severity assessment. Specifically, ISUP Grade 1 corresponds to Gleason 3+3, Grade 2 to 3+4, Grade 3 to 4+3, Grade 4 to Gleason score 8, and Grade 5 to Gleason score 9 or 10, while Grade 0 represents benign samples. The dataset includes a diverse distribution of ISUP grades, with 2,724 slides classified as grade 0 (benign), 2,602 as grade 1, 1,321 as grade 2, 1,205 as grade 3, 1,187 as grade 4, and 1,163 as grade 5. Spanning multiple clinical centers, PANDA ensures a broad range of samples, mitigating center-specific biases.

The Breast Invasive Carcinoma (**TCGA-BRCA**) project is the sub-typing dataset we used. TCGA-BRCA includes two subtypes: Invasive Ductal Carcinoma (**IDC**) and Invasive Lobular Carcinoma (**ILC**). It contains 787 IDC slides and 198 ILC slides from 985 cases.

The primary goal of survival analysis is to estimate the survival probability or survival time of patients over a specific period. Therefore, we used the **TCGA-LUAD** and **TCGA-BRCA** projects to evaluate the model performance for survival analysis tasks. Unlike the diagnosis and sub-typing tasks, the survival analysis datasets are case-based rather than WSI-based. The WSIs of TCGA-BRCA are identical to those used in the sub-typing task but with different annotations. The TCGA-LUAD dataset includes 541 slides from 478 primarily Lung Adenocarcinoma cases.

We randomly split the PANDA dataset into training, validation, and testing sets with a ratio of 7:1:2. Due to the limited data size, the remaining datasets are divided into training and testing sets with a ratio of 7:3. The grading and subtyping tasks use 5 different random seeds to ensure the stability of the results. And because the survival analysis task is more affected by data partitioning, we use 3-fold cross-validation with 3 different random seeds to conduct the experiments.

## A.2  PREPROCESS

Following prior works Lu et al. (2021); Shao et al. (2021); Zhang et al. (2022); Tang et al. (2024), we cropped each WSI into non-overlapping 256x256 patches at $20\times$ magnification. As in CLAM Lu et al. (2021), background regions, including holes, were discarded. The average number of patches is approximately 10,000 for TCGA and 500 for PANDA. To efficiently process the large number of patches, we adopted a traditional two-stage paradigm, employing pre-trained offline models for patch feature extraction. We utilized three state-of-the-art foundation models of varying sizes, pre-trained on WSIs: CHIEF Wang et al. (2024b) (27M), UNI Chen et al. (2024) (307M), and GigaPath Xu et al. (2024) (1134M). Their respective feature dimensions are 768, 1024, and 1536.

## A.3  IMPLEMENTATION DETAILS

For our experiments conducted with a batchsize of 1, which is a conventional approach for methods handling variable sequence lengths like the two-stage methods investigated Lu et al. (2021); Shao et al. (2021); Tang et al. (2024), we consistently employed the Adam optimizer Kingma & Ba (2014). An initial learning rate of $1 \times 10^{-4}$ was used, and this rate was dynamically adjusted during training using the Cosine annealing strategy. To mitigate potential overfitting and ensure robust optimization, we incorporated an early stopping mechanism across all experiments, selecting the model checkpoint that achieved the best performance on the validation metric for final evaluation. For grading tasks, training ran for a maximum of 100 epochs with a patience of 20 (starting from epoch 75). For subtyping, the maximum was 75 epochs with a patience of 20 (starting from epoch 30). For survival analysis, the maximum was 100 epochs with a patience of 10 (starting from epoch 30). To ensure a fair comparison, all baseline methods were tuned following their official guidelines and recommended

hyperparameter search spaces. For experiments involving a batchsize greater than 1, the learning rate was dynamically adjusted to an empirically determined value to achieve optimal performance. Notably, specific models encountered memory limitations on the 3090 GPU when applied to certain large datasets. For instance, training the WiKG aggregator Li et al. (2024c) on the BRCA(Subtyping) dataset required sampling the number of patches down to 1024 per instance to fit into memory. Except for the aforementioned cases, all experiments were performed on NVIDIA 3090 GPUs using unified hyperparameters where applicable.

## B  ADDITIONAL QUANTITATIVE RESULTS

### B.1  MORE DISCUSSION ABOUT HYPERSLIDE

**Effectiveness of the HyperSlide Supervision Signal.** To isolate the contribution of our multi-WSI supervision mechanism, we conducted an ablation study, the results of which are presented in Tab.4. The study demonstrates that the introduction of HyperSlide labels provides a consistent and significant performance uplift across all three downstream tasks. This improvement

Table 4: Impact of HyperSlide supervision.

| Method / Task | Grading | Subtyping | Survival |
|---|---|---|---|
| Random Sampling (RS) | 77.91 | 93.89 | 65.71 |
| RS + HyperSlide | 79.93 | 94.37 | 67.15 |
| Pack (Ours) | 79.02 | 94.06 | 66.15 |
| Pack + HyperSlide (Ours) | **80.19** | **94.21** | **67.50** |

holds true for both the baseline random sampling (RS) packing strategy and our proposed adaptive packing method. For instance, adding HyperSlide to our adaptive packing improved Grading AUC by 1.17, Subtyping AUC by 0.15, and Survival C-Index by 1.35. This underscores the efficacy of using a higher-level, clinically-informed supervision signal to guide the model's learning process on aggregated WSI data.

**Superiority over Soft Labeling.** The design of the HyperSlide label is critical to its success. We further compared our task-specific labeling strategies against a more naive 'mixed soft label' baseline, which simply averages slide-level information without considering task-specific clinical nu-

Table 5: Comparison of task-specific HyperSlide labels and mixed soft-labels.

| Labeling Strategy / Task | Grading | Subtyping | Survival |
|---|---|---|---|
| Mixed Soft Label | 75.50 | 91.31 | N/A |
| Task-specific Label (Ours) | **80.19** | **94.21** | **67.50** |

ances. As shown in Tab.5, our carefully designed labels significantly outperform this simpler approach. The naive method not only yields substantially lower performance on categorical tasks but is also incompatible with event-driven tasks like survival analysis, where label priority is paramount. This result validates our core hypothesis: to effectively learn from multiple WSIs, the generated supervision signal must preserve the essential, task-relevant clinical characteristics of the slide ensemble.

### B.2  MORE DISCUSSION ABOUT BATCHED TRAINING

Table 6: Extended comparison between batched training and gradient accumulation. Batched training not only accelerates training but is essential for the effectiveness of subsequent modules.

| Strategy | Grad. | Sub. | Surv. | TTime |
|---|---|---|---|---|
| *— Without Batched Training —* | | | | |
| Baseline | 73.21 | 93.58 | 65.87 | 12h |
| Accumulation | 72.12 | 93.02 | 64.28 | 12h |
| Accumulation + patch drop | 77.58 | 93.61 | 64.53 | 12h |
| Accumulation + ADS | - | 94.25 | 66.98 | - |
| *— With Batched Training —* | | | | |
| Batched (RS) + patch drop | 77.91 | 93.89 | 65.71 | 2h |
| Batched (RS) + ADS | - | 94.27 | 67.04 | - |
| Batched (RS) + HyperSlide | 79.93 | 94.37 | 67.15 | 2.5h |
| OURS | **80.19** | **94.86** | **68.14** | 4h |

**Batched training is superior to gradient accumulation.** Gradient accumulation is often considered an alternative to batched learning, aiming to simulate larger batch sizes while preserving data heterogeneity. However, our experiments demonstrate its inferiority in CPath tasks that utilize features from foundation models. As detailed in Tab.6, the standard gradient accumulation strategy not only failed to improve training efficiency (maintaining a 12-hour training time) but also consistently underperformed the simple batched training baseline. More importantly, we found that batched training is a crucial prerequisite for unlocking the performance gains from subsequent optimization modules. When combined with techniques like patch dropout or ADS, the batched training approach consistently outperforms its gradient accumulation counterpart. This suggests that the batch-level feature interaction is vital for these modules to function effectively. Furthermore, batched training provides a significant training speedup (up to $8\times$), reducing training time from 12 hours to under 3 hours in most configurations. This efficiency is critical for iterative research and large-scale experimentation. This evidence underscores that the dual benefits of batched training, namely its efficiency and its role as a foundation for advanced modeling, are difficult to replace. In contrast, our proposed packing strategy builds upon the efficiency of batched training. It further enhances data heterogeneity and enables the crucial multi-slide modeling with HyperSlides, achieving the most significant performance gains while maintaining a practical training time.

**Adaptive Packing.** Within our batched training framework, we utilize an adaptive packing strategy. One might consider a simpler approach of packing a fixed number of slides (e.g., pairs) to form each hyperslide. However, our investigation reveals that such a fixed-size strategy is suboptimal. As shown in Tab.7, fixing the pack

Table 7: Comparison of adaptive and fixed-size packing strategies.

| Strategy | Grad. | Sub. | Surv. | Pad. Ratio |
|---|---|---|---|---|
| Fixed-size (n=2) | 79.48 | 94.54 | 67.78 | 54.9% |
| Ours (Adaptive) | **80.19** | **94.86** | **68.14** | **18.5%** |

size to two slides results in a significantly higher padding ratio (54.9% vs. 18.5%). This inefficiency arises because incomplete packs must be padded to a uniform length, leading to wasted computation on uninformative tokens. In contrast, our adaptive packing dynamically fills each hyperslide to its maximum capacity, thereby maximizing token utilization, improving computational efficiency, and yielding superior performance.

**Distinction from Mixup-based Augmentation.** Our HyperSlide methodology is fundamentally distinct from conventional mixup-based data augmentation. The primary distinction lies in the motivation and mechanism. Mixup serves as a regularization technique by creating interpolated training instances and soft labels. In contrast, our goal is to compensate for weak su-

Table 8: Performance comparison with slide-level mixup augmentation.

| Strategy | Grad. | Sub. | Surv. |
|---|---|---|---|
| Slide-level Mixup | 76.25 | 93.23 | N/A |
| Ours (HyperSlide) | **80.19** | **94.86** | **67.50** |

pervision by explicitly modeling inter-slide relationships. We achieve this by constructing clinically meaningful training instances with task-specific macro-labels and a corresponding loss function. This design guides the model to learn from slide ensembles in a clinically relevant manner rather than a purely augmentative one. Furthermore, our approach is length-adaptive, integrating a variable number of slides per pack, unlike typical mixup strategies that operate on fixed pairs. The empirical results in Tab.8 corroborate this conceptual difference, showing that a standard mixup approach fails to match the performance of our purpose-built HyperSlide framework.

## B.3 MORE DISCUSSION ABOUT ADS

While the Attention-driven Downsampler (ADS) module offers benefits in handling redundancy and improving efficiency, its applicability and optimal configuration are subject to certain conditions and data characteristics.

**ADS Behavior at Inference**. For maintaining interpretability at inference time, the ADS module is configured to preserve per-instance information. This is achieved by disabling the pooling operation along the pack dimension ($k$). However, the learned transformations, including the attention score computation via the shallow MLP, the residual weighting ($u_i = h_i + a_i\,h_i$), and the subsequent linear projection ($v_i = u_i\,W^L$), are still applied to each instance. This allows for the analysis of

per-instance attention scores ($a_i$) and transformed features ($v_i$), facilitating post-hoc interpretation of model decisions without reducing the instance count.

**Dependence on Data Redundancy**. The effectiveness of the ADS module is significantly influenced by the inherent redundancy of the input WSI data. For datasets characterized by high tile redundancy, such as TCGA, ADS performs favorably. By reducing the instance count by a factor of $k$, it effectively compresses the representation while discarding redundant

Table 9: Performance of ADS on PANDA.

| Strategy | CHIEF | UNI | GIGAP |
|----------|-------|-------|-------|
| w/ ADS | 76.79 | 78.84 | 77.92 |
| w/o ADS | 76.46 | 80.19 | 80.41 |

or less informative features, leading to computational efficiency and potentially improved signal-to-noise ratio. Conversely, on datasets with intrinsically lower redundancy, like PANDA, applying ADS can be detrimental. In such cases, where a larger proportion of instances may contain clinically relevant information, downsampling can inadvertently discard crucial features, leading to information loss and degraded performance, as shown in Tab. 9. This highlights that the benefits of ADS are most pronounced when applied to data exhibiting substantial spatial redundancy.

**Position of the ADS Module**. The position of the ADS module is a key impact of its efficacy. As shown in Tab. 10, applying the ADS module to the branched feature sets $\mathcal{R}_b$ and $\mathcal{D}_b$ independently (after) yields superior performance over applying it to the combined feature set prior to branching (before). We hypothesize that this is because operating on distinct

Table 10: Performance of ADS on BRCA.

| Strategy | CHIEF | UNI | GIGAP |
|----------|-------|-------|-------|
| before | 90.60 | 94.54 | 94.18 |
| after | 92.38 | 94.86 | 94.86 |

branches enables the ADS module to learn more specialized attention mechanisms. Such specialization allows the module to better capture the unique characteristics and relevant information within each feature set ($\mathcal{R}_b$ and $\mathcal{D}_b$), which might otherwise be obscured or averaged out in a combined representation.

**Computational Cost of the ADS Module**. We evaluate the computational cost and performance impact of incorporating the ADS module. Tab. 11 presents a comparison of computational resources and performance metrics for the model trained with and without the ADS component on the

Table 11: Computational Cost of ADS on BRCA.

| Method | TTime | Memory | FLOPs | AUC |
|--------|-------|--------|-------|-------|
| w/o ADS | 1h | 7G | 49.4G | 94.21 |
| w/ ADS | 1.5h | 13G | 142.1G | 94.86 |

BRCA(Subtyping) dataset. The results show that integrating the ADS module requires additional computational resources. Specifically, training time increases from 1 hour to 1.5 hours, memory usage rises from 7GB to 13GB. Importantly, this investment in computational resources is accompanied by a notable improvement in model performance. The model enhanced with the ADS module achieves an AUC of 94.86%, surpassing the 94.21% obtained by the baseline model. These findings indicate that while the ADS module introduces additional computational requirements, it effectively contributes to a tangible performance gain.

**Attention Mechanism**. The attention mechanism is a critical component of the ADS module. In its absence, the downsampling operation would treat all instances uniformly, assigning equal importance to each. The core contribution of the attention is to compute instance-specific weights, making the feature aggregation process both attention-driven and instance-dependent. This enables the model to selectively focus on and amplify features from the most clinically salient regions. To empirically validate its contribution, we conducted an ablation study by

Table 12: Performance of attention mechanism for subtyping and survival prediction.

| Method | Subtyping (AUC) | Survival (C-Index) |
|--------|-----------------|---------------------|
| w/o Attention | 94.56 | 67.71 |
| w/ Attention | 94.86 | 68.14 |

removing the attention component. As shown in Tab. 12, this resulted in a distinct performance degradation across both subtyping and survival prediction tasks, underscoring the mechanism's importance in learning a meaningful data-driven downsampling policy.

Table 13: Ablation studies on various components of our method. Default settings are marked in gray .

| | Sub. | Surv. |
|---|---|---|
| 2 | 93.87 | 67.03 |
| 3 | 94.40 | 68.14 |
| 4 | 94.86 | 66.89 |
| 5 | 94.32 | 65.74 |
| 6 | 94.60 | 66.03 |

(a) **Downsample ratio** $k$ of ADS.

| | Grad. | Sub. | Surv. |
|---|---|---|---|
| 0.05 | 79.95 | 94.71 | 65.97 |
| 0.1 | 80.15 | 94.78 | 66.42 |
| 0.2 | 80.19 | 94.86 | 67.28 |
| 0.5 | 79.85 | 94.67 | 68.14 |
| 1.0 | 79.94 | 94.56 | 69.31 |

(b) **hyperslide-loss weight** $\lambda$.

| | Grad. | Sub. | Surv. |
|---|---|---|---|
| 30% | 80.01 | 94.58 | 65.94 |
| 40% | 80.19 | 94.86 | 66.69 |
| 50% | 79.52 | 94.77 | 68.14 |
| 60% | 79.04 | 94.86 | 68.23 |

(c) **Branch split ratio** of main branch.

## B.4 ABLATION OF HYPERPARAMETERS

We conduct ablation studies on the hyperparameters related to our method in Tab. 13 and provide the following analysis.

**Dowansmple Ratio of ADS.** The ADS downsampling ratio determines the final number of input instance, demonstrating different characteristics across tasks. For sub-typing tasks, a moderate or smaller number of instances (500–1500) is found to be feasible or even optimal. Consequently, larger downsampling ratios (which result in fewer instances) yielded good performance. However, for the more challenging survival analysis, the number of instances have a more significant impact, with a larger number of instances ($> 2000$) often leading to better performance.

**Weight of Hyperslide Loss.** The influence of varying hyperslide-loss weight on overall optimization is examined. We observe that parameter choices within a certain range consistently provide substantial performance gains, indicating that this parameter is relatively stable. Furthermore, larger ratios are observed to yield better results specifically on the survival analysis task. This improved performance is likely due to the nature of the survival analysis task itself. Given its data volume and inherent difficulty compared to other tasks, the main loss function is more susceptible to overfitting on the training set, making the hyperslide optimization play a more critical role.

**Branch Split Ratio.** The Split ratio controls the proportion of instances in different branches. It is hypothesized that a relatively even distribution would be more conducive to the overall optimization of the dual-branch architecture. Experimental results support this hypothesis, as more uneven division ratios do not yield significant performance gains.

## B.5 ADDITIONAL BENCHMARKING EXPERIMENTS

**Additional Dataset.** To further substantiate the robustness and generalizability of our proposed PackMIL, we extended its evaluation to two additional, widely recognized benchmarks: a computational pathology task for cancer metastasis detection and a medical imaging task outside of pathology for diabetic retinopathy grading. The cancer metastasis detection benchmark integrates the **CAMELYON-16** and **CAMELYON-17** datasets Litjens et al. (2018). These datasets consist of whole-slide images (WSIs) of hematoxylin and eosin (H&E) stained lymph node sections from breast cancer patients. The primary task is to identify the presence of metastatic cancerous tissue within these lymph nodes, a critical step in cancer staging. For the non-pathology benchmark, we utilized a standard **Diabetic Retinopathy (DR) Grading** dataset Li et al. (2019). This dataset contains retinal fundus images and the objective is to classify them into different severity levels of diabetic retinopathy. This task serves to evaluate the model's applicability to broader medical image classification challenges beyond histopathology. We compared PackMIL with several state-of-the-art Multiple Instance Learning (MIL) methods. As shown in Tab. 14, PackMIL demonstrates superior performance on both benchmarks. Specifically, on the CAMELYON cancer metastasis detection task, PackMIL (AB.) achieved an accuracy of 98.42%, outperforming all other methods. Similarly, in the Diabetic Retinopathy Grading task, PackMIL (AB.) obtained the highest score of 61.34%. These results underscore the effectiveness and versatility of our approach across different medical imaging domains.

Table 14: Performance comparison on additional benchmark datasets.

| Method | CAMELYON (UNI) | Diabetic Retinopathy Grading (R50) |
|---|---|---|
| ABMIL | 96.58 | 58.26 |
| DSMIL | 96.44 | 58.03 |
| TransMIL | 96.63 | 60.28 |
| WiKG | 97.08 | 55.84 |
| PackMIL (AB.) | **98.42** | **61.34** |
| PackMIL (DS.) | 97.81 | 60.27 |

**Additional Encoders.** To further assess the versatility and effectiveness of PackMIL, we conducted additional experiments by integrating it with more advanced feature extractors and comparing its performance against , powerful slide-level encoders. We utilized CONCHv1.5 Lu et al. (2024a) as a feature extractor and compared our model with state-of-the-art slide encoders including HIPT Chen et al. (2022b), GigaPath Xu et al. (2024), CHIEF Wang et al. (2024b), and TITAN Ding et al. (2024) on downstream tasks of tumor grading, subtyping, and survival prediction. We reported the training time (TTime) of all methods on PANDA. The results, detailed in Tab. 15, demonstrate that PackMIL consistently enhances performance while maintaining remarkable efficiency. Notably, PackMIL significantly reduces the training time, requiring only 2-3 hours, which is a substantial improvement over the 15 to 75 hours required by other encoders. This highlights PackMIL's ability to effectively aggregate features from various powerful backbones, achieving superior predictive performance with significantly lower computational overhead.

Table 15: Comparative experiments with advanced feature extractors and slide encoders.

| Feature Extractor | Slide Encoder | Grading | Subtyping | Survival | TTime |
|---|---|---|---|---|---|
| HIPT Chen et al. (2022b) | HIPT | $62.10 \pm 1.06$ | $86.93 \pm 4.86$ | $67.82 \pm 9.61$ | 32h |
| GigaPath Xu et al. (2024) | GigaPath | $65.86 \pm 0.77$ | $93.72 \pm 3.42$ | $62.64 \pm 9.33$ | 50h |
| GigaPath Xu et al. (2024) | PackMIL | **$80.41 \pm 0.53$** | $94.86 \pm 3.68$ | $68.03 \pm 9.10$ | 3h |
| CHIEF Wang et al. (2024b) | CHIEF | $67.67 \pm 0.84$ | $91.43 \pm 4.51$ | $67.95 \pm 8.46$ | 15h |
| CONCHv1.5 Lu et al. (2024b) | ABMIL | $66.90 \pm 0.74$ | $95.14 \pm 2.92$ | $68.65 \pm 9.17$ | 14h |
| CONCHv1.5 Lu et al. (2024b) | TITAN | $63.72 \pm 0.76$ | $95.20 \pm 2.92$ | - | 75h |
| CONCHv1.5 Lu et al. (2024b) | PackMIL | $71.72 \pm 0.65$ | **$95.43 \pm 2.63$** | **$70.74 \pm 7.70$** | **2h** |

## B.6 INFERENCE TIME COMPARISON

Tab. 16 presents the inference time with feature input. Since PackMIL only adds the ADS module during inference and retains the simplest MIL inference pipeline, its inference time is nearly identical to the baseline, with less than a 4% loss in inference speed.

Table 16: Performance comparison of various Multiple Instance Learning (MIL) methods, restructured for improved readability on narrower page widths.

| Method | Inference Time (per slide) | FPS |
|---|---|---|
| ABMIL | 0.48625 ms | 2056 fps |
| DSMIL | 1.04138 ms | 960 fps |
| TransMIL | 7.05402 ms | 142 fps |
| DTFD | 3.97953 ms | 251 fps |
| RRT-MIL | 2.43291 ms | 411 fps |
| WIKG | 1.84916 ms | 541 fps |
| PackMIL(ABMIL) | 0.50414 ms | 1984 fps (-4%) |
| PackMIL(DSMIL) | 1.07524 ms | 930 fps (-3%) |
| PackMIL(VITMIL) | 1.36654 ms | 731 fps |
| PackMIL(RRT) | 2.52792 ms | 396 fps (-4%) |

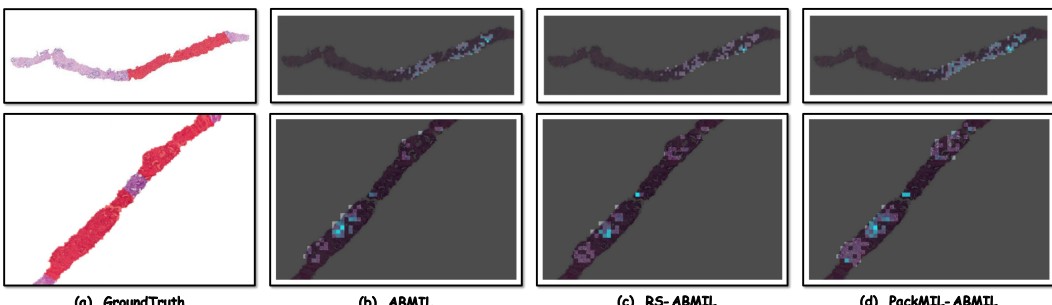

| (a) GroundTruth | (b) ABMIL | (c) RS-ABMIL | (d) PackMIL-ABMIL |

Figure 5: Attention visualization on the PANDA dataset Bulten et al. (2022). The RS strategy, due to its sampling, exhibits limited global attention. With multi-slide supervision via hyperslides and the supplementation of key features, PackMIL demonstrates a more accuracy and comprehensive focus on tumor areas.

### B.7 SUPPLEMENTARY CONFIDENCE INTERVALS

We provide the detailed confidence intervals (CI) for our PackMIL-enhanced methods in Tab. 17, corresponding to the aggregated results presented in the main manuscript.

Table 17: Detailed CI for PackMIL-enhanced methods on all downstream tasks.

| Method | Benchmark | CHIEF | UNI | GIGAP |
|---|---|---|---|---|
| ABMIL +PackMIL | Grading (Acc.) | 76.46±0.61 | 80.19±0.53 | 80.41±0.53 |
| | Sub-typing (AUC) | 92.38±4.12 | 94.86±3.91 | 94.86±3.68 |
| | Survival-BRCA (C-index) | 68.30±8.8 | 68.14±9.8 | 67.04±9.1 |
| | Survival-LUAD (C-index) | 63.72±8.6 | 62.60±8.5 | 61.58±8.7 |
| DSMIL +PackMIL | Grading (Acc.) | 75.84±0.57 | 79.68±0.50 | 79.10±0.55 |
| | Sub-typing (AUC) | 93.01±3.92 | 94.62±3.72 | 94.65±3.14 |
| | Survival-BRCA (C-index) | 69.76±8.5 | 70.00±8.6 | 68.03±9.1 |
| | Survival-LUAD (C-index) | 64.10±8.6 | 62.18±8.7 | 62.44±8.4 |
| TransMIL +PackMIL | Grading (Acc.) | 74.75±0.66 | 78.87±0.57 | 78.88±0.58 |
| | Sub-typing (AUC) | 92.31±4.23 | 94.37±3.83 | 94.12±3.92 |
| | Survival-BRCA (C-index) | 68.08±8.6 | 68.44±8.9 | 66.80±9.1 |
| | Survival-LUAD (C-index) | 64.01±9.1 | 63.61±8.5 | 63.04±8.3 |
| RRTMIL +PackMIL | Grading (Acc.) | 74.63±0.69 | 78.46±0.59 | 78.43±0.60 |
| | Sub-typing (AUC) | 92.43±3.92 | 94.54±3.79 | 94.47±4.01 |
| | Survival-BRCA (C-index) | 68.15±9.1 | 68.73±9.4 | 67.62±9.0 |
| | Survival-LUAD (C-index) | 64.37±8.4 | 62.01±8.6 | 62.79±8.2 |

## C QUALITATIVE ANALYSIS

Here, Fig.5 visualizes the attention scores of different MILs on the PANDA. We suggest that: 1) Due to the data challenges, traditional non-batched training often struggle to achieve efficient and optimal convergence. As a result, the baseline model (ABMIL) exhibits insufficient discriminability and fails to capture some key pathological details. 2) While the simple random sampling (RS) strategy benefits from improved discriminability through batched training, it suffers from feature loss due to sampling and insufficient supervision, resulting in a lack of global attention. 3) In contrast, PackMIL, leveraging pack-based batched training and multi-slide supervision from hyperslides, demonstrates a more accurate and comprehensive focus on tumor areas.

## D  ADDITIONAL METHODOLOGY DESCRIPTION

### D.1  TASK-SPECIFIC HYPERSLIDE LOSS

**Grading task.** This task is modeled as single-label classification, we employ Asymmetric Loss (ASLLoss) Ridnik et al. (2021). This loss function is chosen to effectively address potential class imbalance and encourage the model to focus on correctly identifying positive classes while being less sensitive to negative misclassifications. Given the predicted probability distribution $\mathbf{p} = [p_1, \ldots, p_G]$ for $G$ classes and the corresponding one-hot encoded ground truth labels $\mathbf{y}^{\text{hyper}}$, the loss is computed as:

Table 18: Hyperslide loss ablation on PANDA.

| Strategy | CHIEF | UNI | GIGAP |
|---|---|---|---|
| CE | 76.46 | 80.16 | 80.09 |
| ASL | 76.10 | 80.19 | 80.41 |

$$L_{\text{grade}}(\mathbf{p}, \mathbf{y}^{\text{hyper}}) = -\sum_{i=1}^{G} \mathbf{y}_i^{\text{hyper}}(1 - p_i)^{\gamma_p} \log(p_i) - \sum_{i=1}^{G}(1 - \mathbf{y}_i^{\text{hyper}})p_i^{\gamma_n} \log(1 - p_i), \quad (6)$$

where $\gamma_p \geq 0$ and $\gamma_n \geq 0$ are the focusing parameters for positive and negative samples, respectively.

**Subtyping task.** This task is modeled as multi-label classification with soft targets, we use multi-label Focal Loss Lin et al. (2017). This loss helps mitigate the issue of imbalanced subtype frequencies and focuses the model's attention on harder-to-classify samples. Given the predicted probability vector $\mathbf{p} = [p_1, \ldots, p_C]^\top$ for $C$ subtypes (typ-

Table 19: Hyperslide loss ablation on BRCA.

| Strategy | CHIEF | UNI | GIGAP |
|---|---|---|---|
| BCE | 90.89 | 94.20 | 93.91 |
| FL | 92.38 | 94.86 | 94.86 |

ically obtained via Sigmoid activation) and the corresponding soft ground truth label vector $\mathbf{y}^{\text{hyper}} = [y_1^{\text{hyper}}, \ldots, y_C^{\text{hyper}}]^\top$, the loss is computed as the sum of binary Focal Loss for each class:

$$L_{\text{sub}}(\mathbf{p}, \mathbf{y}^{\text{hyper}}) = \sum_{c=1}^{C} \text{FL}(p_c, y_c^{\text{hyper}}), \quad (7)$$

where, for class $c$, the binary Focal Loss $\text{FL}(p_c, y_c^{\text{hyper}})$ is defined as:

$$\text{FL}(p, y) = -\alpha y(1 - p)^\gamma \log(p) - (1 - y)(1 - \alpha)p^\gamma \log(1 - p). \quad (8)$$

Here, $\alpha \in [0, 1]$ is a weighting factor, and $\gamma \geq 0$ is the focusing parameter.

**Survival analysis task.** The model predicts the hazard of event occurrence over discrete time intervals based on hyper-slice features. The training process employs a custom discrete-time Negative Log-Likelihood (NLL) loss function. Let the follow-up horizon be partitioned into $T$ contiguous, non-overlapping intervals $\{1, \ldots, T\}$. For individual $i$ the model outputs a hazard sequence $\mathbf{h}_i = (h_{i,1}, \ldots, h_{i,T})$ with $h_{i,t} = P(T_i = t \mid T_i \geq t, \mathbf{x}_i)$. The corresponding discrete survival function is:

$$S_{i,t} = \prod_{j=1}^{t}(1 - h_{i,j}), \qquad t = 1, \ldots, T. \quad (9)$$

Denote by $\delta_i \in \{0, 1\}$ the event indicator ($\delta_i = 1$ if the event is observed, 0 if right-censored). Let $k_i$ be the observed event interval if $\delta_i = 1$ and let $k_i^c$ be the last interval in which the subject is known to be at risk when $\delta_i = 0$. We minimise the following per-sample negative log-likelihood

$$\mathcal{L}_i = \delta_i \underbrace{\left[-\log h_{i,k_i} - \log S_{i,k_i}\right]}_{\mathcal{L}_{\text{event}, i}} + (1 - \delta_i)(1 - \alpha)\underbrace{\left[-\log S_{i,k_i^c}\right]}_{\mathcal{L}_{\text{cens}, i}}, \quad (10)$$

where $\alpha \in [0, 1]$ down-weights the censored component. The mini-batch loss is

$$\mathcal{L} = \frac{1}{B}\sum_{i=1}^{B}\mathcal{L}_i. \quad (11)$$

Hazards are produced by a sigmoid layer, and survival probabilities are obtained by the cumulative product $S_{i,t} = \prod_{j \leq t}(1 - h_{i,j})$. When indices are stored in a 1-based convention, the hazard of the

$k$-th interval must be accessed at position $k-1$ of the zero-based tensor, whereas survival $S_{i,k}$ is accessed at position $k$ after prefix-padding with an initial 1. For censored observations an optional indicator $\mathbf{1}_{k_i = k_i^c}$ can be applied if the censoring interval exactly coincides with a potential event interval; otherwise every censored instance contributes its survival term.

## D.2 Construction of Isolated Mask

As described in the method section, all auxiliary masks fall into two functional groups: *aggregation-oriented* masks, which constrain feature interaction inside each pack, and *classification-oriented* masks, which identify the source bag of every valid token for the downstream classifier. We first introduce the shared primitives and then derive both groups. Let $P_p = \{h_k\}_{k \in \mathcal{I}_p}$ be the $p$-th pack, padded to length $L$ and assembled from $B$ bags. For positions $j = 1, \ldots, L$ we define

$$(\mathbf{m}_p)_j = \mathbb{I}\{j \text{ indexes a real feature}\}, \qquad \mathbf{m}_p \in \{0,1\}^L, \tag{12}$$

$$(\mathbf{b}_p)_j = \begin{cases} \beta_k, & j \text{ contains } h_k, \\ 0, & j \text{ is padding}, \end{cases} \qquad \mathbf{b}_p \in \{0, \ldots, B\}^L, \tag{13}$$

where $\beta_k \in \{1, \ldots, B\}$ is the global bag index of feature $h_k$. By construction $(\mathbf{m}_p)_j = 1$ iff $(p-1)L + j \leq M$, with $M$ the total number of tokens before packing.

**Aggregation-oriented masks.** These masks guarantee that feature aggregation never crosses bag boundaries or attends to padding. We construct a binary feature mask and an attention mask for each pack. Binary feature mask $\mathbf{M}_p \in \{0,1\}^{L \times B}$, indicating which bag each feature belongs to:

$$(\mathbf{M}_p)_{j,b} = (\mathbf{m}_p)_j \cdot \mathbb{I}\{(\mathbf{b}_p)_j = b\}, \quad \text{for } j = 1, \ldots, L, \; b = 1, \ldots, B. \tag{14}$$

An attention mask $\mathbf{A}_p \in \{-\infty, 0\}^{L \times L}$ to enforce intra-bag attention and prevent attention to padding:

$$\mathbf{A}_p = -\infty \left[ \mathbf{1}_{L \times L} - (\mathbf{m}_p \mathbf{m}_p^\top) \odot \mathbf{E}_p \right], \quad (\mathbf{E}_p)_{ij} = \mathbb{I}\{(\mathbf{b}_p)_i = (\mathbf{b}_p)_j\}. \tag{15}$$

Here, $\mathbf{1}$ is the all-ones matrix, $\odot$ denotes element-wise multiplication, and $\mathbf{E}_p$ checks if features $i$ and $j$ belong to the same original bag. Equivalently, $\mathbf{A}_p$ can be visualized in explicit $L \times L$ block-matrix form:

$$\mathbf{A}_p = \begin{pmatrix} \mathbf{0}_{n_1 \times n_1} & -\infty \, \mathbf{1}_{n_1 \times n_2} & \cdots & -\infty \, \mathbf{1}_{n_1 \times n_B} & -\infty \, \mathbf{1}_{n_1 \times n_0} \\ -\infty \, \mathbf{1}_{n_2 \times n_1} & \mathbf{0}_{n_2 \times n_2} & \cdots & -\infty \, \mathbf{1}_{n_2 \times n_B} & -\infty \, \mathbf{1}_{n_2 \times n_0} \\ \vdots & \vdots & \ddots & \vdots & \vdots \\ -\infty \, \mathbf{1}_{n_B \times n_1} & -\infty \, \mathbf{1}_{n_B \times n_2} & \cdots & \mathbf{0}_{n_B \times n_B} & -\infty \, \mathbf{1}_{n_B \times n_0} \\ -\infty \, \mathbf{1}_{n_0 \times n_1} & -\infty \, \mathbf{1}_{n_0 \times n_2} & \cdots & -\infty \, \mathbf{1}_{n_0 \times n_B} & -\infty \, \mathbf{1}_{n_0 \times n_0} \end{pmatrix}, \tag{16}$$

where

$$n_b = \sum_{j=1}^{L} \mathbb{I}\{(\mathbf{b}_p)_j = b\}, \quad b = 1, \ldots, B, \qquad n_0 = L - \sum_{b=1}^{B} n_b$$

counts tokens from bag $b$ (and $n_0$ counts padding tokens) within pack $p$. $\mathbf{0}_{a \times a}$ is the zero matrix, and $\mathbf{1}_{a \times b}$ is the all-ones matrix.

**Classification-oriented masks.** After aggregation, we must (i) keep only valid tokens and (ii) reveal their bag labels to the classifier. Both goals are achieved with

$$\mathbf{v}_p = \mathbf{m}_p, \qquad \text{(valid-token indicator)}, \tag{17}$$

$$\mathbf{c}_p = \mathbf{b}_p, \qquad \text{(bag-label vector)}. \tag{18}$$

Here $\mathbf{v}_p$ filters out padding positions before the prediction head, whereas $\mathbf{c}_p$ routes each remaining token to the correct bag-level logit.

Aggregation-oriented masks act inside the encoder to enforce intra-bag interactions, while classification-oriented masks operate at the output stage to attach each valid token to its original bag. Together they preserve bag integrity and enable efficient batched processing without introducing extra parameters.

### D.3 Dynamic Pack Length Adaptation

While the pack operation utilizes a fixed length $L$ for efficient batch processing, the actual number of instances sampled from each bag ($|\tilde{\mathcal{R}}_b|$ and $|\tilde{\mathcal{D}}_b|$) can vary significantly due to stochastic sampling and the diverse sizes of original WSIs. To accommodate this inherent variability, particularly when a mini-batch

Table 20: Ablation on Dynamic Pack Length.

| Strategy | Grad. | Sub. | Surv. |
|---|---|---|---|
| Fixed Length | 79.69 | 94.42 | 67.72 |
| Dynamic Length | 80.19 | 94.86 | 68.14 |

contains bags that yield a large number of sampled instances, we incorporate a dynamic pack length adaptation mechanism. Before packing the instances for a given branch (main or residual) within a mini-batch, we assess the maximum sampled sequence length from any single bag in that batch. Specifically, we check if $\max_b |\tilde{\mathcal{R}}_b|$ (for the main branch) or $\max_b |\tilde{\mathcal{D}}_b|$ (for the residual branch) exceeds the current pack length $L$. If this condition is met, the pack length for that specific branch and mini-batch is dynamically doubled to $2L$. This dynamic doubling occurs at most once per branch per mini-batch processing step, effectively setting an upper bound of $2L$ on the pack length. This adaptation ensures that sampled instances from bags with particularly large retained or discarded sets are less likely to be fragmented across numerous packs, leading to more efficient packing and potentially better representation within packs for such cases.

## E Additional Related Works

### E.1 Pack-based Batched Training

Training on variable-length sequences has traditionally relied on padding shorter sequences and applying masks to ignore padded tokens, ensuring uniform batch shapes at the cost of wasted computation Krell et al. (2021). To reduce this overhead, dynamic batching (length-based bucketing) groups sequences of similar lengths per batch, greatly minimizing padding requirements Żelasko et al. (2025). Modern transformer architectures further exploit attention masks to handle padding, and recent work goes beyond simple padding by packing multiple sequences into one longer sequence with special separators and adjusted position indices Krell et al. (2021). Such sequence packing techniques, originally used in large-scale NLP pre-training, can double throughput by eliminating pad tokens Kosec et al. (2021) while maintaining model fidelity via careful masking to prevent cross-sequence attention. For example, packing algorithms in BERT pre-training combine several short sentences into a single 512-token input, yielding 2× speedups with negligible accuracy loss Kosec et al. (2021). In computer vision, analogous ideas enable variable-resolution training. NaViT avoids fixed-size resizing by treating images as sequences of patches and packing arbitrary-resolution inputs, improving efficiency in large-scale image–text pre-training without sacrificing performance Dehghani et al. (2023). Other works dynamically reduce sequence length during processing, such as Token Merging (ToMe) merges redundant tokens in ViTs to halve the token count on the fly, boosting throughput 2× for large models with minimal accuracy drop Bolya et al. (2022). RNN-based systems commonly use packed sequences to skip computation on padded timesteps, and Transformer-based LLMs and vision models use padding masks or adaptive token pruning to similar effect. In CPath, where WSI yields a bag of thousands of instances, efficient batching is critical. However, the data characteristics in CPath render the direct application of the aforementioned strategies non-trivial. Approaches focused on packing short sequences provide limited benefits, while sampling long sequences risks significant information loss. Effectively adapting these efficient training paradigms to CPath is thus a key challenge. Our proposed pack-based framework addresses this by incorporating a residual branch to more effectively packing these variable-length long sequences, aiming to mitigate these limitations.

### E.2 More about Batchsize in Computational Pathology

As elaborated in the *Related Work* section, current slide-level MIL methods often train with batchsize of 1. Conversely, when explicit patch-level annotations are available for segmentation or detection tasks, researchers commonly employ moderate to large batch sizes by independently processing uniformly-sized patches extracted from WSIs, enabling stable training and efficient convergence Ciga et al. (2021); Graham et al. (2019). These models independently process uniformly-sized patches extracted from WSIs, leveraging moderate to large batch sizes to facilitate stable training and

efficient convergence Ciga et al. (2021); Graham et al. (2019). Conversely, when explicit patch-level annotations are available for segmentation or detection tasks, researchers typically form batches at the patch level. They independently process uniformly-sized patches from WSIs using moderate to large batchsizes, facilitating stable training and efficient convergence Ciga et al. (2021); Graham et al. (2019). Such patch-centric batching, however, presents a discrepancy with the holistic slide assessment in clinical workflows.

## F    LIMITATION & BROADER IMPACTS

This work revisiting data challenges in computational pathology and, by considering these challenges, proposes a pack-based MIL training framework. However, the primary limitation of this method is the significant challenge in implementing batched training of complex MIL models based on packs. While we have implemented with commonly used models such as ABMIL, TransMIL, and DSMIL, constructing the required masks for some more complex model structures remains challenging. Furthermore, the current hyperslide training is highly specific to downstream tasks. Designing a more general training objective is a key focus of our future work. Beyond these limitations, this work holds significant potential to advance key healthcare tasks such as cancer diagnosis and prognosis. The significant performance improvements demonstrated in this work, particularly when leveraging foundation model features, hold potential to benefit and inspire the development of more accurate state-of-the-art algorithms in the clinical scenario.

## G    STATEMENT ON THE USE OF LARGE LANGUAGE MODELS

Throughout the preparation of this manuscript, we utilized large language models to enhance the quality of the text. Specifically, we employed Google's Gemini for tasks related to language refinement, including correcting grammar and spelling, improving sentence clarity, and ensuring a consistent academic tone. The core scientific contributions, including the formulation of the problem, the proposed methodology, the design and execution of experiments, and the interpretation of results, are entirely the work of the authors. All text generated or modified by the LLM was meticulously reviewed, edited, and revised by the authors to ensure it accurately reflects our original ideas and findings. The authors bear full and final responsibility for all content presented in this paper.

## H    REPRODUCIBILITY STATEMENT

To ensure the reproducibility of our research, we provide a comprehensive set of resources. The complete source code for our proposed framework, along with the implementations of all baseline models used for comparison, is available in an anonymous repository at https://anonymous.4open.science/r/PackMIL-A320. To further facilitate direct replication of our experimental results, we will make the pre-extracted features for all datasets publicly available upon acceptance. Furthermore, a complete Docker container specifying the full computational environment and all dependencies will be released to eliminate any potential for environment-related discrepancies. Detailed descriptions of our data preprocessing pipeline, from whole-slide images to feature extraction, are provided in Appendix A. All hyperparameters and experimental configurations are fully documented in Appendix B.4. We believe these resources will enable the community to readily verify our findings and build upon our work.

## I    DATA AVAILABILITY STATEMENT

The PANDA dataset (CC-BY-4.0) is available at https://panda.grand-challenge.org/.

All TCGA datasets can be found at https://portal.gdc.cancer.gov/.

The CAMELYON dataset is available at https://camelyon17.grand-challenge.org/.

The Diabetic Retinopathy Grading dataset is available at https://github.com/nkicsl/DDR-dataset.

