# OpenReview forum: "Revisiting Data Challenges of Computational Pathology: A Pack-based Multiple Instance Learning Framework"
_ICLR.cc/2026/Conference — ICLR 2026 Conference Withdrawn Submission_

### Official Review · Reviewer_WAoG · 2025-10-22

**Soundness:** 2
**Presentation:** 2
**Contribution:** 2
**Rating:** 4
**Confidence:** 5

**Summary:**

This paper proposes an approach to support batching and random sampling for multiple instance learning (MIL). Importantly, this approach is agnostic to the specific MIL architecture, meaning it can be added to any MIL method. This is particularly important given the vast array of MIL methods proposed or in existence.

In standard practice for computational pathology (CPath), images are of heterogeneous size. As stated by the paper, their length can range from 400-20,000 patches at the standard size of 20x magnification and 256x256 pixels. This presents a challenge for practitioners wishing to use a non-unit batch size. The standard solution to this is to apply random sampling of a fixed number of patches from each WSI. As mentioned by the paper, this sampling process itself can also improve generalizability by acting as a regularizer. The paper finds that random sampling is not consistently beneficial across all tasks, potentially due to the loss of information from discarded patches. To maintain the gradient among discarded patches, the paper proposes a 2-arm pack-based approach. In the first model component, they randomly sample a fixed number of patches for batching and combine them into smaller packs of length L. A downsampling module is then applied to perform weighted pooling, reducing the pack length by a factor of k in order to condense redundant patches. An MIL-based aggregator is then applied to each of these packs. In the second arm, the discarded patches across all slides in the batch are aggregated into a hyperslide in order to preserve the gradient and stabilize training. MIL is applied to both the hyperslide and the pooled packs. Loss is a weighted sum of the pack predictions and hyperslide. Given that the hyperslide can consist of patches with different slide-level labels, this work proposes a task-specific pseudolabel approach, depending on whether the task is grading (weighted average), subtyping (multilabel), or survival (min time to event). Their approach leads to consistent improvements in model performance across a range of tasks.

Overall, the approach is interesting and relevant to the CPath community. However, its presentation and experiments have substantial room for improved clarity. I would be willing to revise my score if these issues are addressed.

**Strengths:**

- This approach tackles an unaddressed challenge in CPath of enabling batched inputs. While random sampling with batched inputs is standard practice in the community, the benefits are not well explored or discussed. The proposed approach can function with any number of MIL methods and tasks.
- The tasks range from subtyping, grading, and survival analysis. Its benefit is shown across eight MIL methods.
- The approach is novel and mostly reasonable

**Weaknesses:**

- Many claims are unsubstantiated or not well substantiated (e.g Lines 49-50, 134-135, 178-179). In particular, a central claim of the paper is that batched inputs can substantially improve performance (Lines 138-139). While this work indicates how performance changes (both runtime and accuracy) with random sampling vs. random sampling + batched inputs in Table 3, the change from RS alone vs. RS with batching is very small. To show the effect of batching alone, performance should be shown with zero-padding. Similarly, Lines 369-370 and Line 414 claim that the benefits of random sampling is primarily due to batched processing and reduced input redundancy. Again, there is little presented evidence of how beneficial the batching is towards final performance, and random sampling does not necessarily decrease input redundancy, as this sampling process is input-agnostic. In Line 417, it is claimed that the significant gains in performance is due to batched training, yet the gains from adding batching to random sampling is small according to Table 3.
- It is not clear why sequence length heterogeneity would affect training stability when performing random sampling (Lines 371-373). Could it be because the positive instances are being discarded when the sequence length is too large? Consider explaining this or other more understandable hypotheses.
- The ADS module needs additional details. Please show patches or heatmaps showcasing how the pooling is performed in order to confirm or deny that morphologically-similar patches are being condensed.
- The deltas in table 2 are incorrect. For instance, DSMIL with packmil is 69.76 while base DSMIL is 65.81, but delta is listed as 2.9. Please double check values across all tables.
- The writing quality should be substantially improved for both grammar and clarity, particularly in the introduction, related works, and methods.
- Table and figure captions are vague. Please define any acronyms present in the figures (e.g RS=random sample in Figure 2).

The methods section should be substantially restructured for clarity.
- Please provide a description of the intuition behind what a pack is near line 184-185)
- Please move the definition of the pack operation near equation 2.
- Please clarify early on around lines 183-191 that random sampling is only done during training rather than inference.
- Describe the goal of forming packs. It should be clear that mixing instances from different bags is a byproduct of packing with batched inputs, which you resolve with masking.
When introducing the mask for instances from different bags within the same pack, please first explain why there might be a mix of bag instances per pack.
- Describe what a hyperslide represents near its introduction in line 225 and its purpose. The goal is to retain the gradient among discarded slides by joining all the discarded patches, using a pseudolabel which represents this collection of patches to resolve the issue of joining instances from different slides.
- Ensure that you distinguish between features and instances. Unless I am mistaken, in Line 231, I believe M is the total number of *instances*, even though the text states that it is the features. This mix up is also present in line 232 and 235, and potentially elsewhere in the text.
- Rather than applying MIL on the full hyperslide, why not apply MIL with masking to each bag, and using the bag-level label rather than a hyperslide pseudolabel? Would this not be more memory-efficient for O(n^2) approaches such as self-attention, while also reducing label noise? This specific reasoning does not need to be included within the text, but intuition behind why a hyperslide might be beneficial should be described when the residual branch is introduced in lines 225-229.
- Please explain instance unshuffle more clearly (Lines 252-254)
- In line 265, it is claimed that ADS pooling prioritizes regions of high clinical relevance. Elsewhere, it is described as an approach to reduce redundant information. However, this downsampling is performed either by random or max pooling according to equation 4. Please provide evidence that max pooling is indeed able to prioritize these key features or patches.

- The hyperslide label for grading is unintuitive to me. This label is currently obtained by a an average of bag labels within the minibatch, weighted by the number of patches in the hyperslide. However, grade is typically determined as the *highest* grade observable within a slide. That is, if grade=6 is observed anywhere within an image, then the entire biopsy is labeled as grade=6. Consequently, this design decision to use a weighted average for grading requires either additional justification or empirical evidence for performance gain.

- Figure 5: performance with batch size=1 needs to be included for all plots. This is a critical component and the most important benchmark value.
- Line 314-15: Please explain the rule. This claim is an overgeneralization as well, as only one “regular rule for scaling” has been investigated.
- Lines 365-366: Citations are needed for this claim. See [1] and [2].

[1] Chen, S. et al. Benchmarking Embedding Aggregation Methods in Computational Pathology: A Clinical Data Perspective.
[2] Shao, D. et al. Do Multiple Instance Learning Models Transfer? In Forty-second International Conference on Machine Learning, June 2025

- Please clarify why Table 15 is missing survival results on TITAN, and indicate whether these results against slide encoders are with finetuning or linear probe. FEATHER [2] benchmarking should also be added, as it is also based on ABMIL and Conch v1.5.

**Questions:**

- Why is it necessary to allow packing instances from different bags? Intuitively, I imagine that forcing each pack to consist of instances from a single bag would also be effective, by padding with duplicate instances (as is done in TransMIL’s nystrom mechanism) or adding zero-padding, without substantially increasing the number of padding necessary compared to the current approach.

- Is $a_i$ normalized in 251?

- I can’t seem to find details on whether random or max pooling in equation 4 is used for experiments.

- PANDA performance should be reported as quadratic weighted kappa, following standard convention.
- Figure 2: Please add error bars. Describe the number of patches used in random sample.
- Consider renaming R_b and D_b. It is confusing that R represents “retained” instances, but discarded instances are sent to a Residual branch.
- Ensure that all tables and figures in the appendix are mentioned in the main manuscript.

---

### Official Review · Reviewer_yT4d · 2025-10-30

**Soundness:** 3
**Presentation:** 2
**Contribution:** 2
**Rating:** 4
**Confidence:** 3

**Summary:**

The paper proposes a new training strategy for Whole Slide Image (WSI)-level prediction in Computational Pathology (CPATH). In standard CPATH pipelines, a WSI is subdivided into tiles, each tile is encoded using a pretrained foundation model, and the resulting tile embeddings are aggregated - typically via attention-based Multiple Instance Learning (MIL) - to produce a slide-level representation trained to predict the slide-level target (e.g. grading, subtype or survival). The authors argue that this approach is computationally challenging because WSIs contain highly variable and often extremely large numbers of tiles. Existing methods address this by random tile subsampling during training or by using a batch size of one; both approaches can be rightfully considered to be limited in some way: a batch size of 1 can lead to problems in training and tile subsampling might lead to an omission of the relevant tiles.

To overcome these limitations, the authors introduce a pack-based training framework in which each mini-batch includes tile embeddings from multiple WSIs. For each WSI, tile embeddings are first randomly partitioned into a retained set (R) and a discarded set (D). Both sets are further subsampled ($\tilde{R}$, $\tilde{D}$). These embedding subsets are then grouped into fixed-size “packs”, such that each pack contains tile encodings originating from multiple WSIs. The downstream network is trained on these packs to make two kinds of predictions: (1) standard WSI-level predictions (in which case the slide origin is used to separate the contribution of different slides) from the main branch ($\tilde{R}$) and (2) “hyperslide” predictions, which correspond to aggregated targets derived from all WSIs composing the pack in the residual branch ($\tilde{D}$). As slides can have different sizes (sequence lengths), zero-padding is used to complement the packs. The authors perform benchmarking experiments against random subsampling and single-slide mini-batch training across multiple foundation models, aggregation strategies, and datasets covering four prediction tasks (breast cancer survival, lung cancer survival, breast cancer grading, and breast cancer subtyping). Results show that the proposed approach yields a consistent, albeit moderate, performance improvement over existing baselines.

**Strengths:**

- The paper addresses an interesting question, as it is likely that aggregation and subsampling strategies are not yet optimally solved by current approaches.
- The method consistently outperforms competing strategies

**Weaknesses:**

1. The authors state that “training by sampling … can lead to a loss of data heterogeneity and important features” (line 076). But then, the proposed method also heavily relies on subsampling, including stochastic subsampling. So, this problem does not seem to be solved by the proposed approach.
2. There are many unclear statements, and some formal issues. For instance, the representation $z = \Gamma_{\theta}(\sum_i h_i)$ does not seem correct, as the most popular approach relies on an attention mechanism that weights the tile encodings. Another example: Figure 5 has no caption. Figure 5.c has no y-axis values. Furthermore, it is unclear why only large batch-sizes are shown in Figure 5, when the major claim is that batch-size=1 is the most popular choice (which is not true in the case of random subsampling). More generally, I do not see what we can learn from Figure 5.
3. In Table 1, we do see the performances of the trainings with batch-size=1 with standard deviation. First, the standard deviations are surprisingly small, and it is not clear how they were calculated. Second, for the sampling results, no CI is given. However, we need them to assess the statistical significance of the reported results.
4. Zero-padding is likely to introduce a heavy bias. Indeed, smaller WSI will generate many zero-valued pack contributions; the network might therefore learn shortcuts, if the size is related to targets.
5. Not in all publications batch-size = 1 is used. ("Consequently, mainstream slide-level MIL methods typically adopt a batchsize of 1 Jaume et al. (2024) l130)". For instance, Jaume et al. (2024) do not use batch size = 1 and actually invest the effect of batch size in this article (see Supplementary Figure 3).
6. I was wondering about the target aggregation for multi-slide supervision. For instance, the aggregated grading target might not correspond to any of the original grades. This might be problematic if the grade does not correspond to a continuum of visual features.

**Questions:**

1. What happens if you train only with slide-level supervision (no hyperslide prediction), or vice versa? How much does each branch contribute?
2. How can you rule out bias introduced by zero padding, which can be seen as a proxy for slide size?
3. What is the precise motivation for partitioning tiles into retained (R) and discarded (D) sets? Did you evaluate how sensitive the method is to the split ratio or to reusing the same subsets across epochs?
4. Is there a theoretical reason to expect that mixing tiles from different slides (in packs) improves generalization, or is it primarily empirical?

---

### Official Review · Reviewer_5iVk · 2025-10-31

**Soundness:** 2
**Presentation:** 2
**Contribution:** 2
**Rating:** 4
**Confidence:** 4

**Summary:**

This paper addresses three core data challenges in computational pathology (CPath) involving whole slide images (WSIs): high data heterogeneity (sequence lengths varying from 200 to 200K), high redundancy, and limited supervision. To tackle these issues, the authors propose a pack-based MIL framework (PackMIL) with three key components: (1) a packing mechanism that converts multiple variable-length feature sequences into fixed-length packs to enable batched training while retaining heterogeneity via masks; (2) a residual branch that constructs a "hyperslide" from discarded features of multiple slides, paired with task-specific labels to provide multi-slide supervision and mitigate feature loss; and (3) an attention-driven downsampler (ADS) to compress redundant features in both branches. Extensive experiments on datasets including PANDA (prostate cancer grading), TCGA-BRCA (cancer subtyping/survival analysis), and TCGA-LUAD (survival analysis) show that PackMIL achieves up to 11% accuracy improvement on PANDA and reduces training time to only 12% of traditional methods (≈8× speedup).

**Strengths:**

1. **Comprehensive addressing of core CPath challenges**: Unlike existing works that only tackle individual issues (e.g., random sampling for batch training but losing heterogeneity), PackMIL systematically integrates packing, residual hyperslide, and ADS to simultaneously alleviate heterogeneity, redundancy, and limited supervision—filling a gap in holistic solutions for CPath data challenges.
2. **Significant efficiency and performance gains**: The framework demonstrates tangible improvements in both training efficiency (e.g., reducing TransMIL’s training time on PANDA from 55 hours to 6.5 hours) and predictive accuracy (e.g., +7.0% accuracy for ABMIL on PANDA grading), supported by rigorous experiments with 1000 bootstrap iterations and multiple random seeds.

**Weaknesses:**

1. **Incomplete coverage of related works on efficiency-accuracy tradeoffs**: The authors overlook several recent MIL methods that also balance computational performance and accuracy, such as Attention-Challenging MIL (ACMIL) [1], Context-Aware MIL (CAMIL) [2], Attention Entropy Maximization (AEM) [3], and Hierarchical Distillation MIL (HDMIL) [4]. Additionally, the paper fails to compare with algorithms published after 2025 (e.g., HDMIL and AEM, both 2025), which address similar efficiency or supervision challenges. Omitting these works and recent advances weakens the paper’s ability to position PackMIL within the current state-of-the-art and highlight its unique contributions.
2. **Lack of discussion on WSI-specific packing-related works**: The paper does not introduce or compare with existing research on WSI packing, such as the "Slide Packing" approach proposed in [5] (Aswolinskiy et al., 2025). The absence of this comparison leaves readers unaware of PackMIL’s innovations relative to prior packing-based ideas for CPath.
3. **Unvalidated claim on extreme sequence length variations**: The authors emphasize "significant sequence length variations (200 to 200K)" as a key challenge, but such extreme variations typically occur across datasets (e.g., TCGA-BRCA with ~60K sequence length vs. PANDA with ~1K, as noted in the paper). However, all experiments are conducted on individual datasets (e.g., PANDA alone, TCGA-BRCA alone), without testing PackMIL on scenarios with cross-dataset extreme length variations. This gap raises doubts about whether the framework can handle the very heterogeneity it claims to address.

[1] Yunlong Zhang, Honglin Li, Yunxuan Sun, Sunyi Zheng, Chenglu Zhu, and Lin Yang. Attentionchallenging multiple instance learning for whole slide image classification. In European Conference on Computer Vision, pp. 125–143. Springer, 2024.

[2] Olga Fourkioti, Matt De Vries, and Chris Bakal. CAMIL: Context-aware multiple instance learning for cancer detection and subtyping in whole slide images. In The Twelfth International Conference on Learning Representations, 2024. URL https://openreview.net/forum?id= rzBskAEmoc.

[3] Yunlong Zhang, Zhongyi Shui, Yunxuan Sun, Honglin Li, Jingxiong Li, Chenglu Zhu, and Lin Yang. Aem: Attention entropy maximization for multiple instance learning based whole slide image classification. International Conference on Medical Image Computing and Computer Assisted Intervention, 2025.

[4] Jiuyang Dong, Junjun Jiang, Kui Jiang, Jiahan Li, and Yongbing Zhang. Fast and accurate gigapixel pathological image classification with hierarchical distillation multi-instance learning. In Proceedings of the Computer Vision and Pattern Recognition Conference, pp. 30818–30828, 2025.

[5] Aswolinskiy W, van der Post RS, Campora M, Baronchelli C, Ardighieri L, Vatrano S, van der Laak J, Munari E, Simons M, Nagtegaal I, Ciompi F. Attention-based whole-slide image compression achieves pathologist-level pre-screening of multi-organ routine histopathology biopsies. Modern Pathology. 2025 Jun 23:100827.

**Questions:**

1. Regarding the incomplete related works (Weakness 1): Have the authors considered the MIL methods ACMIL [1], CAMIL [2], AEM [3], and HDMIL [4]? If so, why were these works not included in the comparison, and how does PackMIL’s efficiency-accuracy tradeoff differ from these methods (e.g., HDMIL’s hierarchical distillation for fast gigapixel image classification)?
2. Regarding extreme sequence length variations (Weakness 3): Do the authors plan to conduct additional experiments on cross-dataset scenarios with extreme sequence length variations to validate PackMIL’s ability to handle such heterogeneity? If not, could the authors explain why experiments on individual datasets are sufficient to demonstrate PackMIL’s effectiveness against the claimed 200–200K length variation challenge?

---

### Official Review · Reviewer_a7ra · 2025-10-31

**Soundness:** 2
**Presentation:** 1
**Contribution:** 2
**Rating:** 2
**Confidence:** 4

**Summary:**

The authors propose a pack-based MIL framework for WSI classification in histopathology. This addresses the problem of variable bag sizes for different WSIs in MIL-based frameworks, which induces inefficiencies by constraining MIL methods to be trained with batch size 1 or by resampling to fixed-size bags. PackMIL also introduces an attention-driven downsampler module to downsample bag size by a fixed factor. It also introduces an hyperslide that aggregates discarded patches from multiple slides and creates an aggregate label following one of several strategies depending on the specific task. The method is demonstrated on top of several MIL aggregators: ABMIL, DSMIL, TransMIL, RRTMIL, using UNI, CHIEF or GigaPath patch encoders. It is evaluated on PANDA (cancer grading), TCGA-BRCA (survival analysis and cancer sub-typing) and TCGA-LUAD (survival analysis), showing performance improvements over a random sampling strategy where all input bags are resized to a fixed length.

**Strengths:**

Significance: The proposed pack-based MIL strategy addresses a real difficulty stemming from the variable size of bags in MIL-based WSI classification approaches.

Originality: This type of hyperslide strategy could potentially help improve diversity and act as a form of augmentation that increases artificially the number of slides, with positive effects on performance (but this is inconclusive from the paper).

**Weaknesses:**

Clarity and quality: In its current version, the paper leaves many questions unanswered, both regarding methodology and results (see questions below).
- The ADS mechanism is still unclear in its mechanism and impact. (residual weight a_i, instance unshuffle mechanism and pooling)
- The hyperslide label creation also raises questions in some cases.
- The packing process raises many unanswered questions (large vs. small bag contributions, what happens when a bag size exceeds L, etc.)
- How does the packing strategy integrates with the different MIL paradigms? For TransMIL, this is probably similar to [1], but for other MILs specific steps might be required to integrate masking in the MIL approach? The integration of the packing with specific MILs should be described.

Effectiveness: The ablation study shows only minor gains from two of the stated contributions: hyperslide and ADS. (less than 1 point in the relevant metrics) Is this improvement statistically significant?

Positioning: The paper is not positioned with previous work on packing such as [1].

[1] Dehghani et al. Patch n' Pack: NaViT, a Vision Transformer for any Aspect Ratio and Resolution, NeuRIPS 2023

**Questions:**

1. ADS: How is k chosen and does the pooling take into account the spatial configuration of patches in the slide? How does the instance unshuffle work exactly?

2. Can authors clarify the rationale behind the residual contribution a_i h_i, and how a_i is computed?

3. beta_n is defined but not used in the paper if I am not mistaken. Can the authors clarify?

4. The authors mention the high sequence length heterogeneity, from 200 to 200K. Since all slides are downsampled by an identical factor, this heterogeneity persists after packing. Since packs have a fixed length L, this should yield extreme imbalance between large and small bags in a pack. Can authors please clarify? The paper mentions: "we enforce a minimum number of patches
per slide in each pack". This could have a substantial effect but this is not elaborated further upon. What is the effect of this minimum number hyperparameter?

5. There are several steps where a given bag is split or downsampled. But the information relevant for the diagnosis can sometimes be highly localized within a given bag. It is possible that the patches relevant to a cancer diagnosis or subtype is moved to the hyperslide, or removed when downsampling, which would introduce inconsistency between the slide label and what is left of the bag. Can the authors comment on this?

6. Similarly, the hyperslide label is created by aggregating individual slide labels according to their contribution to the hyperslide. But the part of the WSI that is moved to the hyperslide could potentially be inconsistent with the original slide label.

7. "To ensure statistical robustness, we perform 1000 bootstrap iterations and repeat experiments across multiple random seed": When and how many random seeds and runs are used, vs. when are boostrapped test sets used? Bootstrapping the test set instead of training several times the MIL is likely to underestimate uncertainty.

8. "We report the mean and 95% confidence interval for all metric" : Intervals are not reported for the proposed PackMIL. Is the reported number an average over several runs or the result of a single run? There is large variability across runs in MIL methods.

9. Tables 1 and 2 include other MIL methods (CLAM, DTFD, etc.) on top of which PackMIL was not tested. Could the authors clarify why? Or why these baselines are included in the table, since we do not know if combining them with PackMIL would improve over them.

10. Are improvements over the random sampling baseline statistically significant?

11. Does the PackMIL strategy integrate transparently with arbitrary MIL methods? This is not elaborated upon, but the fact that patches from several slides end up in the same "sample" might call for adjustments to all specific MIL implementations to account for the masking?

12. What is the effect of lambda on the final performance?

13. Is it possible for a bag to be longer than L and split over several "samples"? If so, are the resulting "samples" processed entirely independently afterwards by the MIL method?

---

### Note · Authors · 2025-11-20

I have read and agree with the venue's withdrawal policy on behalf of myself and my co-authors.